# FANCM limits ALT activity by restricting telomeric replication stress induced by deregulated BLM and R-loops

Bruno Silva[1], Richard Pentz[1], Ana Margarida Figueira[1], Rajika Arora[1], Yong Woo Lee[1], Charlotte Hodson[2], Harry Wischnewski[3], Andrew J. Deans [2,4] & Claus M. Azzalin [1]

Telomerase negative immortal cancer cells elongate telomeres through the Alternative Lengthening of Telomeres (ALT) pathway. While sustained telomeric replicative stress is required to maintain ALT, it might also lead to cell death when excessive. Here, we show that the ATPase/translocase activity of FANCM keeps telomeric replicative stress in check specifically in ALT cells. When FANCM is depleted in ALT cells, telomeres become dysfunctional, and cells stop proliferating and die. FANCM depletion also increases ALT-associated marks and de novo synthesis of telomeric DNA. Depletion of the BLM helicase reduces the telomeric replication stress and cell proliferation defects induced by FANCM inactivation. Finally, FANCM unwinds telomeric R-loops in vitro and suppresses their accumulation in cells. Overexpression of RNaseH1 completely abolishes the replication stress remaining in cells codepleted for FANCM and BLM. Thus, FANCM allows controlled ALT activity and ALT cell proliferation by limiting the toxicity of uncontrolled BLM and telomeric R-loops.

[1] Instituto de Medicina Molecular João Lobo Antunes (iMM), Faculdade de Medicina da Universidade de Lisboa, Lisbon 1649-028, Portugal. [2] Genome Stability Unit, St. Vincent's Institute of Medical Research, Fitzroy 3065 VIC, Australia. [3] Institute of Biochemistry (IBC), Eidgenössische Technische Hochschule Zürich (ETHZ), Zürich 8093, Switzerland. [4] Department of Medicine (St Vincent's Hospital), University of Melbourne, Fitzroy 3065 VIC, Australia. Correspondence and requests for materials should be addressed to C.M.A. (email: cmazzalin@medicina.ulisboa.pt)

Telomere shortening must be counteracted in immortal cells, including the large majority of cancer cells, to avoid senescence or death[1]. Approximately 90% of human cancers have reactivated the reverse transcriptase telomerase, which adds newly synthesized telomeric repeats to the 3′ end of linear chromosomes[2,3]. About 10% of immortal cancer cells are telomerase-negative and replenish telomeres using the so-called Alternative Lengthening of Telomeres (ALT) pathway[4]. In humans, ALT was reported in tumors of mesenchymal or epithelial origin, including osteosarcomas, liposarcomas, glioblastomas, astrocytomas, and bladder carcinomas as well as in in vitro immortalized cell lines[4–8].

Molecular features considered markers for ALT comprise: (i) telomeres of heterogeneous lengths at different chromosome ends, including telomeres much longer than average telomeres in telomerase-positive cells[8]; (ii) elevated levels of the telomeric long noncoding RNA (lncRNA) TERRA[9–13]; (iii) clustering of multiple telomeres into ALT-associated PML bodies (APBs), nuclear structures containing promyelocytic leukemia protein (PML), telomeric factors such as TRF1, TRF2 and RAP1, TERRA, and DNA repair factors such as RAD51, RAD52, Replication Protein A (RPA), Brca1, and Bloom (BLM) and Werner helicases[11,14–19]; (iv) abundant extrachromosomal telomeric repeats (ECTRs) comprising double-stranded (ds) circles (t-circles), partially single-stranded (ss) circles (C- and G-circles) and linear dsDNA[20–23]; (v) recurrent mutations of the Alpha Thalassemia/Mental Retardation Syndrome X-Linked (ATRX) gene[12].

Multiple DNA metabolism pathways collaborate to maintain telomeres in ALT cells. Break-induced replication (BIR) is active at ALT telomeres in the G2 phase of the cell cycle, and is stimulated by DSBs experimentally induced using the telomere-tethered DNA endonuclease TRF1-FokI[24,25]. ALT BIR requires POLD3 and POLD4, two regulatory subunits of DNA polymerase delta[24,25]. Conservative mitotic DNA synthesis (MiDAS) was also documented in human ALT cells[26]. ALT MiDAS is stimulated by replication stress and requires RAD52[26]. Finally, clustering of ALT telomeres within APBs is promoted by RAD51-dependent long-range movements, which are also stimulated by TRF1-FokI-induced DSBs[27]. Telomere movements may promote efficient homology searches and telomere synthesis, although both ALT BIR and MiDAS are independent of RAD51[25,26].

A common notion deriving from all this work is that a sustained physiological damage must be maintained at ALT telomeres to promote telomere elongation. This is consistent with the presence of replication stress and DNA damage markers in APBs[11,14–18]. The triggers of this damage remain unclear, although RNA:DNA hybrids (R-loops), G-quadruplexes and oncogene expression were proposed as candidates[11,26]. This scenario implies that telomeric damage levels be maintained within a specific threshold that is high enough to trigger DNA synthesis-based repair, yet not too high to induce cell death. Consistently, telomeric R-loops (telR-loops) formed by TERRA and telomeric DNA activate replication stress at ALT telomeres, and their levels are tightly controlled by the endoribonuclease RNaseH1[11,28]. When RNaseH1 is depleted, excessive replication stress rapidly leads to abundant telomere free chromosome ends (TFEs) and increased C-circles. Conversely, RNaseH1 overexpression causes progressive TFE accumulation, likely due to inefficient de novo synthesis of telomeric DNA[11]. The DNA damage signaling kinase ATM- and Rad3-Related (ATR) and the annealing helicase SWI/SNF-related matrix-associated actin-dependent regulator of chromatin subfamily A-like protein 1 (SMARCAL1) were also reported to restrict replicative stress at ALT telomeres[29,30].

The Fanconi anemia, complementation group M (FANCM) ATPase/translocase is a component of the Fanconi Anemia (FA) complex, where it supports efficient FANCD2 ubiquitination

upon stalling of replication forks by physical impediments including DNA crosslinks[31]. Independently of the FA complex, FANCM remodels replication forks, recruits DNA repair factors at damage sites, suppresses meiotic crossovers and facilitates ATR checkpoint activation[32–35]. Moreover, the ATPase/translocase activity of FANCM resolves RNA:DNA hybrids in vitro and that R-loops accumulate genomewide in FANCM-deficient cells[36]. We hypothesized that FANCM suppresses replication stress at ALT telomeres, and while this work was in progress, a report from Pan et al.[17] confirmed our hypothesis. The authors showed that, in ALT cells, FANCM allows efficient progression of the replication fork through the telomeric tract, and depletion of FANCM induces telomeric replication stress[17]. The same study also reported that FANCM depletion leads to accumulation of BLM and Brca1 at ALT telomeres and that codepletion of FANCM with Brca1 or BLM is lethal[17].

Here we show that FANCM depletion in ALT cells causes robust telomere replication stress and damage, activation of ATR signaling, nearly complete abrogation of proliferation, and cell death. In FANCM-depleted ALT cells telomeric ssDNA, ECTRs and mitotic DNA threads accumulate. Moreover, features of ALT activity including APBs and DNA synthesis in G2/M augment when FANCM is depleted. An ATPase/translocase inactive variant of FANCM fails to revert telomeric replication stress and APB accumulation in cells depleted for endogenous FANCM. Finally, FANCM resolves telR-loops in vitro and restricts them in cells, and the replicative stress induced by FANCM depletion is completely averted by simultaneous codepletion of BLM and overexpression of RNaseH1. We propose that FANCM keeps replicative stress and ALT in check by assuring regulated BLM activity and resolving telR-loops.

## Results

**FANCM supports viability of ALT cells.** We depleted FANCM in several ALT (U2OS, HuO9, Saos2 and WI-38 VA13) and telomerase-positive (Tel+; HeLa, HOS, HT1080 and SKNAS) cells using short interference RNAs (siRNAs) against two sequences from FANCM coding region (siFa and siFb). Non-targeting siRNAs were used as controls (siCt). Two days after transfection, nearly complete depletion of FANCM protein was detected by western blot in Fa- and Fb-transfected cells, with the exception of siFb-transfected SKNAS cells, where about 10% of the protein remained (Fig. 1a). Fluorescence-activated cell sorting (FACS) of ethanol-fixed, propidium iodide (PI)-stained cells revealed that FANCM-depleted ALT cells, but not Tel+, accumulated in G2/M phase (Fig. 1b, c; Supplementary Fig. 1A). The clonogenic potential of ALT cells was largely abolished upon transfection of FANCM siRNAs, while the one of Tel+ cells remained essentially unaffected (Fig. 1d, e). For colony formation experiments, cells were transfected only once with siRNAs before seeding and colonies were counted at least 8 days later. Hence, the antiproliferative effects exerted by FANCM depletion on ALT cells are fast and irreversible. Cell growth analysis upon prolonged siRNA treatment showed that FANCM-depleted U2OS cells were quickly eliminated from the population, while HeLa cells continued to grow although at lower rates (Fig. 1f). Finally, FANCM-depleted U2OS cells, but not HeLa cells, started to be permeable to PI already after 3 days of siRNA treatment denoting cell death (Supplementary Fig. 2A). No major changes in PARP1 cleavage were detected in the same cells (Supplementary Fig. 2B). Thus, FANCM depletion causes aberrant accumulation of ALT cells in G2/M phase, followed by PARP1-independent cell death.

Our data indicate that FANCM is essential for cell-cycle progression and viability in ALT cells. This is different to what

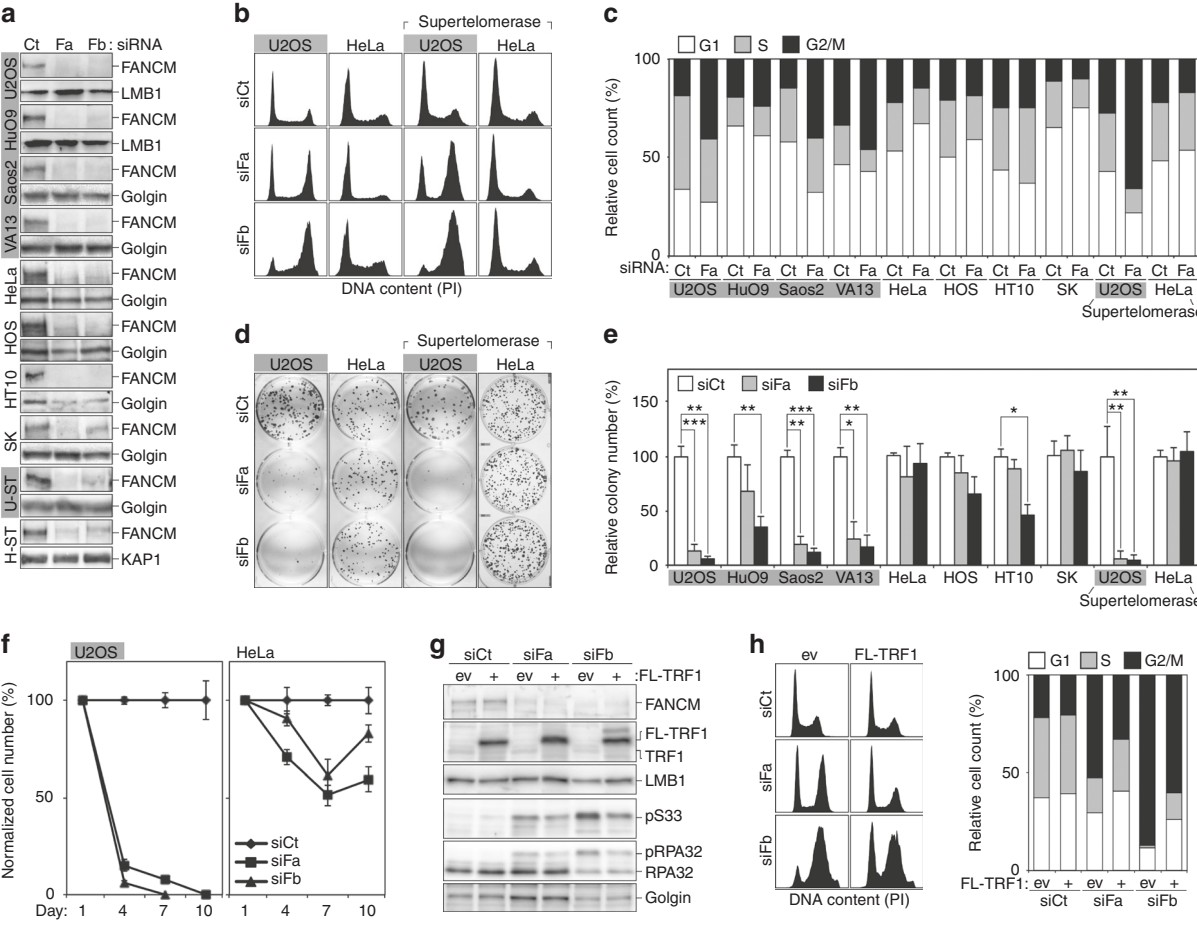

**Fig. 1** FANCM supports normal cell-cycle progression and proliferation of ALT cells. **a** Western blot analysis of FANCM protein levels in ALT and Tel+ cells transfected with anti-FANCM siRNAs (siFa and siFb) or with control siRNAs (siCt). ALT cells (gray background) are: U2OS, HuO9, Saos2 and WI-38 VA13 (VA13); Tel+ cells are: HeLa, HOS, HT1080 (HT10) and SKNAS (SK). U-ST and H-ST are supertelomerase U2OS and HeLa cells, respectively. Proteins were extracted 48 h after transfection. Lamin B1 (LMB1), Golgin 97 and KAP1 serve as loading controls. **b** Examples of FACS profiles of the indicated siRNA-transfected cells stained with propidium iodide (PI). Cell counts (y axis) are plotted against PI intensity (x axis). Cells were harvested 48 h after transfection. **c** Quantifications of experiments as in (**b**). The graph shows the percentage of cells in G1, S and G2/M phases from one representative experiment. **d** Examples of colony formation assays with the indicated siRNA-transfected cells. **e** Quantifications of experiments as in (**d**). The graph shows colony numbers relative to siCt-transfected samples. Bars and error bars are means and SDs from three independent experiments. P values were calculated with a two-tailed Student's t test. *P < 0.05, **P < 0.005, ***P < 0.001. **f** Growth curves of U2OS and HeLa cells transfected with the indicated siRNAs every 3 days. Cell numbers are expressed relative to siCt-transfected cells. Data points and error bars are means and SDs from three independent experiments. **g** Western blot analysis of U2OS cells infected with retroviruses expressing Flag-tagged TRF1 (FL-TRF1) or with empty vector (ev) control retroviruses. Five days after infections cells were transfected with the indicated siRNAs and harvested 48 h later. pS33: RPA32 phosphorylated at serine 33, pRPA32: phosphorylated RPA32. LMB1 and Golgin serve as loading controls. **h** Examples of FACS profiles of cells as in (**g**). The graph on the left shows the percentage of cells in G1, S and G2/M phases from one representative experiment. Source data are provided as a Source Data file

was observed previously[17]. It is possible that less efficient protein depletion obtained by Pan and colleagues or retained expression of crucial FANCM splice variants (possibly including cell-type specific ones that are not reported in public databases) left residual amounts of FANCM protein sufficient to sustain cell proliferation. Less sensitive cell viability assays might also have underestimated the effects of FANCM depletion in the previous study.

**Telomeric replication stress sensitizes ALT cells to FANCM depletion.** Several features of ALT cells could explain their sensitivity to FANCM depletion: absence of telomerase activity, very long telomeres, ATRX inactivation, and sustained telomeric replication stress. We ectopically expressed the catalytic (hTERT) and RNA (hTR) subunits of telomerase in U2OS and HeLa cells to generate supertelomerase cells[37]. Overexpression of hTERT and hTR was confirmed by quantitative RT-PCR (Supplementary

Fig. 3A). As expected, HeLa supertelomerase cells had much longer telomeres than HeLa control cells; U2OS supertelomerase cells had reduced TFE frequencies, while the incidence of under-replicated, fragile telomeres (TFs) remained unchanged (Supplementary Fig. 3B)[11,37]. FANCM depletion inhibited cell proliferation and led to G2/M accumulation in U2OS supertelomerase cells, but not in HeLa supertelomerase cells (Fig. 1b–e). Moreover, HeLa cells treated with the telomerase inhibitor BIBR 1532 [38] did not accumulate in G2/M when depleted for FANCM (Supplementary Fig. 4A). We then code-pleted FANCM and ATRX in HeLa cells and did not observe the accumulation of G2/M cells (Supplementary Fig. 4B). Thus, the presence of ultra-long telomeres or the absence of active telomerase or ATRX alone do not explain the sensitivity of ALT cells to FANCM depletion.

We then overexpressed the shelterin factor TRF1 in U2OS cells by retroviral infection, as this treatment halves the incidence of

FTs[39]. FANCM depletion in TRF1 overexpressing cells still led to G2/M accumulation, yet less severely than in cells infected with empty vector (ev) retroviruses (Fig. 1g, h). However, HeLa cells depleted for TRF1 using an siRNA previously shown to induce telomere fragility[39] did not accumulate in G2/M when codepleted for FANCM (Supplementary Fig. 4C). Similarly, FANCM-depleted HeLa cells did not show an altered cell-cycle distribution when treated with the replication stress inducer hydroxyurea (HU) followed by block release (Supplementary Fig. 4D). Hence, telomeric replication stress contributes to the sensitivity of ALT cells to FANCM depletion; nevertheless, telomeric or generalized replication stress alone are not sufficient to sensitize non-ALT cells to FANCM depletion.

**FANCM suppresses telomeric replication stress in ALT cells.** To test the involvement of FANCM in telomere stability, we performed indirect immunofluorescence (IF) using antibodies against TRF2 combined with antibodies against RPA32 phosphorylated at Serine 33 (pS33) or p53 binding protein 1 (53BP1). RPA32 is phosphorylated at serine 33 during S phase by ATR upon replication fork stalling[40]; 53BP1 forms foci at dysfunctional telomeres that have activated either ATR, or the other DNA damage signaling kinase ataxia-telangiectasia mutated (ATM), or both[41,42]. Within 48 h of transfection, pS33 and 53BP1 accumulated at telomeres in FANCM-depleted ALT cells (Fig. 2a, b) forming the so-called telomere dysfunction-induced foci (TIFs)[41]. FANCM depletion did not induce TIF formation in Tel+ cells (Fig. 2a, b). Accumulation of pSer33 and 53BP1 outside of telomeres was negligible in all FANCM-depleted cell lines (Fig. 2a). FANCM-mediated suppression of telomere instability is likely to be direct, because the protein associated with telomeric DNA in chromatin immunoprecipitation (ChIP) experiments (Fig. 2c, d). FANCM also immunoprecipitated with the abundant, genomewide-spread Alu repeat DNA (Fig. 2c, d), indicating that the protein is not exclusively associated with telomeres. This is consistent with the reported localization of FANCM to cellular chromatin fractions[43].

Western blot analysis confirmed that FANCM depletion causes pS33 accumulation and revealed phosphorylation of the other ATR target checkpoint kinase 1 (CHK1) in U2OS but not HeLa cells (Fig. 2e). The ATM target KRAB domain-associated protein 1 (KAP1) was not phosphorylated in any of the tested cell lines (Fig. 2e). Moreover, pS33 accumulation was weakened in FANCM-depleted U2OS cells overexpressing TRF1 (Fig. 1g), while telomerase inhibition, ATRX or TRF1 depletion and HU treatment did not promote pS33 accumulation in FANCM-depleted HeLa cells (Supplementary Fig. 4A-D). Actually, pS33 failed to accumulate efficiently in HeLa cells depleted for FANCM and treated with HU, consistent with a role for FANCM in supporting activation of the canonical ATR-dependent intra S-phase checkpoint[32]. We propose that FANCM deficiency in ALT cells activates a specific ATR-dependent signaling cascade, which is not fully identical to the one triggered by generalized replication stress and stems at least partly from excessive telomeric replication stress. Such ATR response likely provokes the observed G2/M arrest and cell death.

**FANCM suppresses ALT features.** In our IF images, TRF2 foci in FANCM-depleted ALT cells are both larger and brighter than in control cells (Fig. 2a). To confirm that this was not simply due to increased TRF2 at telomeres, we subjected siRNA-transfected U2OS interphase cells to DNA fluorescence in situ hybridization (FISH) using telomeric probes, and measured the number and area of telomeric foci. We controlled for possible secondary effects related to cell-cycle stage by arresting siCt-transfected cells

at the G2/M border with the cyclin-dependent kinase 1 (CDK1) inhibitor RO-3306[44] (Fig. 3a, b; Supplementary Fig. 1B). The overall number of telomeric foci decreased upon FANCM depletion (Fig. 3c, d), while their area distribution was broader, with slightly increased frequencies of very small foci (S in Fig. 3c–e) and substantially increased frequencies of very large foci (L in Fig. 3c–e). RO-3306-treated cells also had less telomeric foci than control cells (Fig. 3d), likely due to clustering of ALT telomeres in G2 [19,45]. However, the increase in very small and very large foci was more pronounced upon FANCM depletion than RO-3306 treatment (Fig. 3c–e). Approximately 60% of FANCM-depleted cells had at least five large foci, vs. approximately 10 and 15% of untreated or RO-3306-treated siCt-transfected cells, respectively (Fig. 3f).

We then analyzed the localization of PML, RAD51 and POLD3 at telomeres by combining PML and RAD51 IF with telomere FISH, and double IF for POLD3 and RAP1. We observed increased telomeric localization of all three factors in FANCM-depleted U2OS cells, without obvious increase in PML, POLD3 and RAD51 total protein levels (Figs. 3a, 4a, b). RO-3306 treatment did not substantially affect the number of telomeric PML and RAD51 foci, while it increased the one of telomeric POLD3 foci yet less importantly than FANCM depletion (Fig. 4a, b). Moreover, we incubated cells treated as above with the thymidine analog 5-Ethynyl-2′-deoxyuridine (EdU) for 2.5 h, and performed telomere FISH combined with EdU detection to visualize newly synthesized telomeric DNA (Fig. 4a). To exclude S phase cells, we only scored cells showing a punctuate EdU staining and with not more than 25 EdU foci. FANCM depletion increased the incidence of telomeric EdU foci, as it did RO-3306 treatment albeit to lower extents (Fig. 4a, b).

We conclude that FANCM depletion exacerbates ALT activity as shown by robust telomere clustering within large APBs containing PML, RAD51 and POLD3, and increased synthesis of telomeric DNA outside of S phase. FANCM depletion also generates short telomeric species, possibly representing ECTRs (see below). G2/M arrest alone cannot explain the aberrantly elevated ALT features observed in FANCM-depleted cells.

**FANCM suppresses telomeric ssDNA and ECTRs in ALT cells.** We performed in-gel telomere restriction fragment (TRF) analysis of genomic DNA from ALT (U2OS and WI-38 VA13) and Tel+ (HOS and HeLa) cells harvested 48 h after siRNA transfection. Blots were hybridized with telomeric oligonucleotides of either 5′-TTAGGG-3′ or 5′-CCCTAA-3′ repeats. When hybridization was performed under native (nondenatured) conditions, we observed increased C-rich telomeric ssDNA of very diverse lengths in FANCM-depleted ALT cells (Fig. 5a, upper panel). Conversely, a decrease of G-rich ssDNA was observed in correspondence of the bulk of telomeres, likely due to shortening of the G-overhang (Fig. 5a, lower panels). For both probes, a fraction of the signal was in the gel wells, possibly corresponding to ssDNA exposed from molecules with significant secondary structures (Fig. 5a). We did not observe alteration of telomeric ssDNA in Tel + cells (Fig. 5a). Hybridization of the same gels in denatured conditions using a long telomeric probe (Telo2 probe) revealed no appreciable alteration of telomere length in FANCM-depleted cells (Fig. 5a).

We then dot-blotted genomic DNA from cells as above and hybridized it under native conditions to telomeric oligonucleotides, followed by denaturation and hybridization with an Alu repeat, as a control for total DNA loaded. This experiment confirmed that FANCM-depleted ALT cells contain more telomeric C-rich ssDNA than siCt-transfected cells (Fig. 5b). As previously reported[11], depletion of RNaseH1 in U2OS cells also

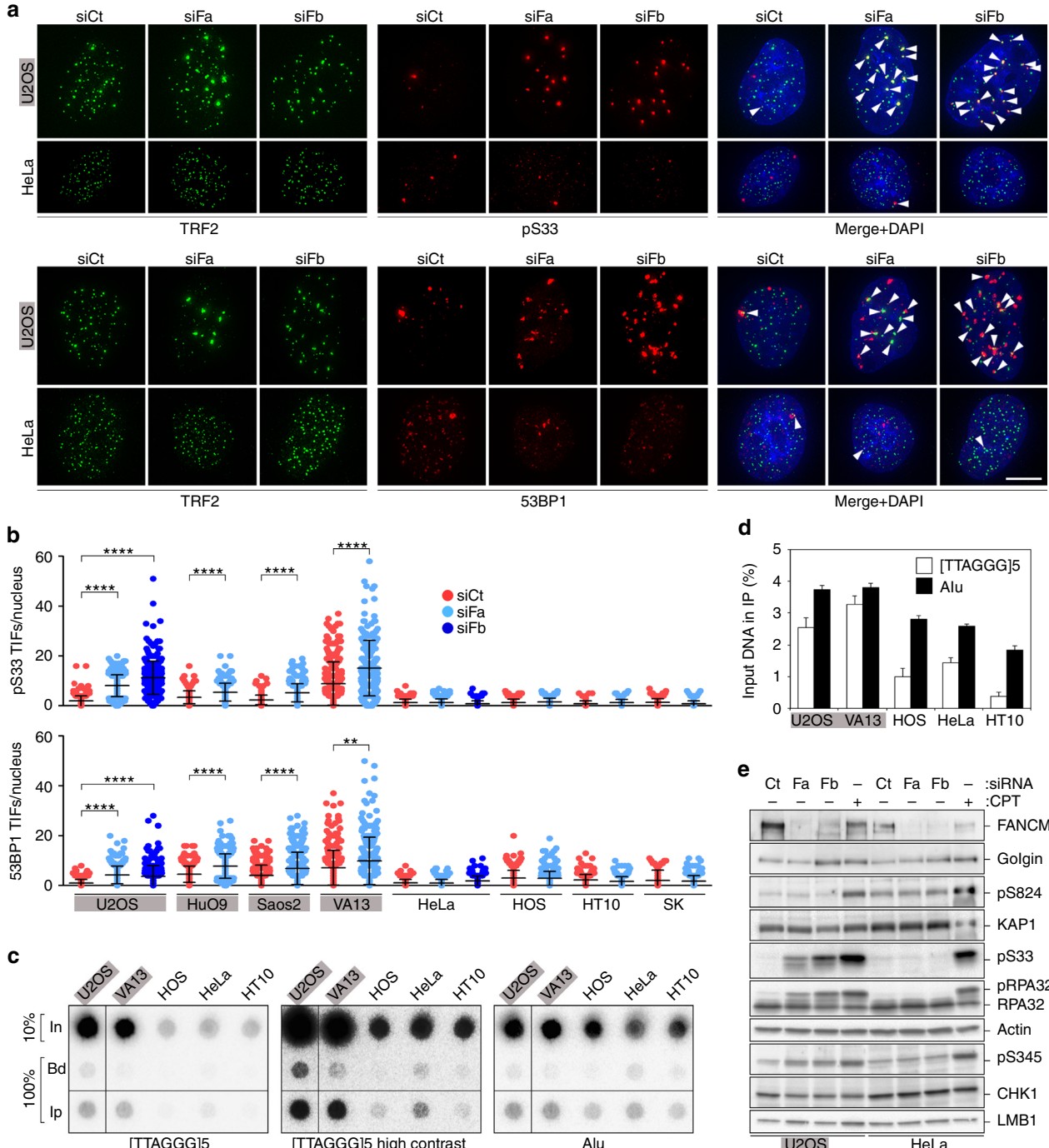

**Fig. 2** FANCM suppresses telomeric DNA damage and localizes to telomeres in ALT cells. **a** Examples of pS33 or 53BP1 immunostaining (red) combined with TRF2 immunostaining (green) on U2OS and HeLa cells transfected with the indicated siRNAs and harvested 48 h after transfection. In the merge panel, DAPI-stained DNA is also shown (blue). Arrowheads point to pS33 and 53BP1 TIFs. Scale bar: 10 μm. **b** Quantifications of numbers of TIFs per nucleus in experiments as in (**a**) performed on the indicated cell lines. ALT cells are on a grey background. Each dot represents an individual nucleus. A total of at least 196 nuclei from three independent experiments were analyzed for each sample. Bars and error bars are means and SDs. $P$ values were calculated with a Mann−Whitney $U$ test. **$P$ < 0.005, ****$P$ < 0.0001. **c** Dot-blot hybridization of endogenous FANCM ChIPs in the indicated cell lines using radiolabeled oligonucleotides comprising telomeric G-rich repeats or Alu repeats. A high contrasted image is shown to facilitate visualization of the telomeric signal for Tel+ cells. In Input, Bd only beads control, Ip anti-FANCM immunoprecipitation. **d** Quantifications of experiments as in (**c**). Signals are graphed as the fraction of In found in the corresponding Ip samples, after subtraction of Bd-associated signals. Bars and error bars are means and SDs from three independent experiments. **e** Western blot analysis of DNA damage activation in the indicated siRNA-transfected cells. Proteins were extracted 48 h after transfection. Untransfected cells treated with camptothecin (CPT) were included to control for antibody specificity. pS824: KAP1 phosphorylated at serine 824, pS345: CHK1 phosphorylated at serine 345. Beta Actin and LMB1 serve as loading controls. Source data are provided as a Source Data file

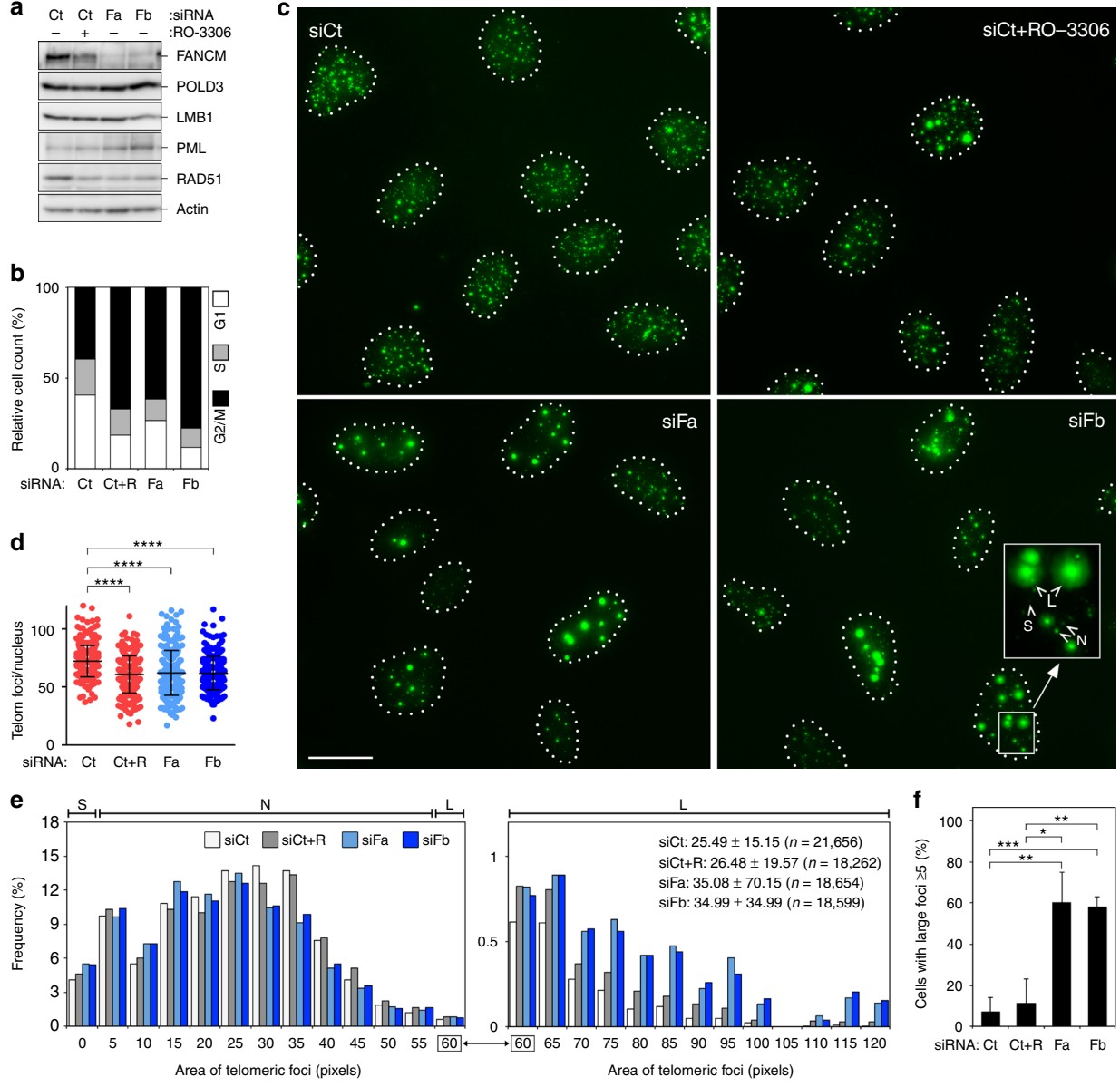

**Fig. 3** FANCM depletion alters telomeric DNA in interphase U2OS cells. **a** Western blot analysis of FANCM protein levels in U2OS cells transfected and treated with RO-3306 as indicated. Cells were harvested 48 h after transfection. POLD3, RAD51 and PML levels were also analyzed. Beta Actin and LMB1 serve as loading controls. **b** Quantifications of FACS profiles of cells as in (**a**) stained with PI. The graph shows the percentage of cells in G1, S and G2/M phases from one representative experiment. R: RO-3306. **c** Examples of telomeric FISH on interphase cells as in (**a**). Telomeric DNA is shown in green, nuclear outlines are shown by dotted lines. S small foci, N normal foci, L large foci. Scale bar: 20 μm. **d** Quantifications of numbers of telomeric foci per nucleus in experiments as in (**c**). Each dot represents an individual nucleus. A total of at least 300 nuclei from three independent experiments were analyzed for each sample. Bars and error bars are means and SDs. *P* values were calculated with a Mann−Whitney *U* test. **e** Area distribution of telomeric foci areas in experiments as in (**c**). 3D images were sum projected and areas of individual nuclear FISH signals were measured using DAPI staining to identify nuclei (not shown). A total of at least 300 nuclei from three independent experiments were analyzed for each sample. Areas of telomeric foci (in pixels) are binned into 25 intervals of 5-pixel width (*x* axis; numbers indicate bin centers) and plotted against frequencies (*y* axis; %). S small foci (0−2.5 pixels), N normal foci (2.5−57.5 pixels), L large foci (57.6−125.5 pixels). The distribution of large foci is represented in the right graph using a smaller *y* axis scale to facilitate visualization. **f** Quantification of cells with at least five Large (L) foci in experiments as in (**c**). *P* values were calculated with a two-tailed Student's *t* test. *$P < 0.05$, **$P < 0.005$, ***$P < 0.001$, ****$P < 0.0001$. Source data are provided as a Source Data file

increased telomeric C-rich ssDNA, albeit at lower levels than FANCM depletion (Fig. 5b). No major difference in total telomeric DNA was detected using dot-blot hybridization of denatured DNA for U2OS, HOS and HeLa cells (Fig. 5b). An increase in total C-rich telomeric DNA was observed in FANCM-depleted WI-38 VA13 cells (Fig. 5b).

We then performed phi-29-mediated C-circle assays[46] using DNA from ALT and Tel+ cells and found a remarkable increase

in C-circles in ALT cells depleted for FANCM (Fig. 5c). Accumulation of ECTRs, likely to correspond partly but not exclusively to C-circles, was also detected in FANCM-depleted U2OS cells using two-dimensional gel electrophoresis (Fig. 5d). Metaphase chromosome FISH of FANCM-depleted U2OS cells showed abundant extrachromosomal telomeric signals, probably corresponding to ECTRs, and DNA threads extending from the termini of single chromosomes, or bridging two independent

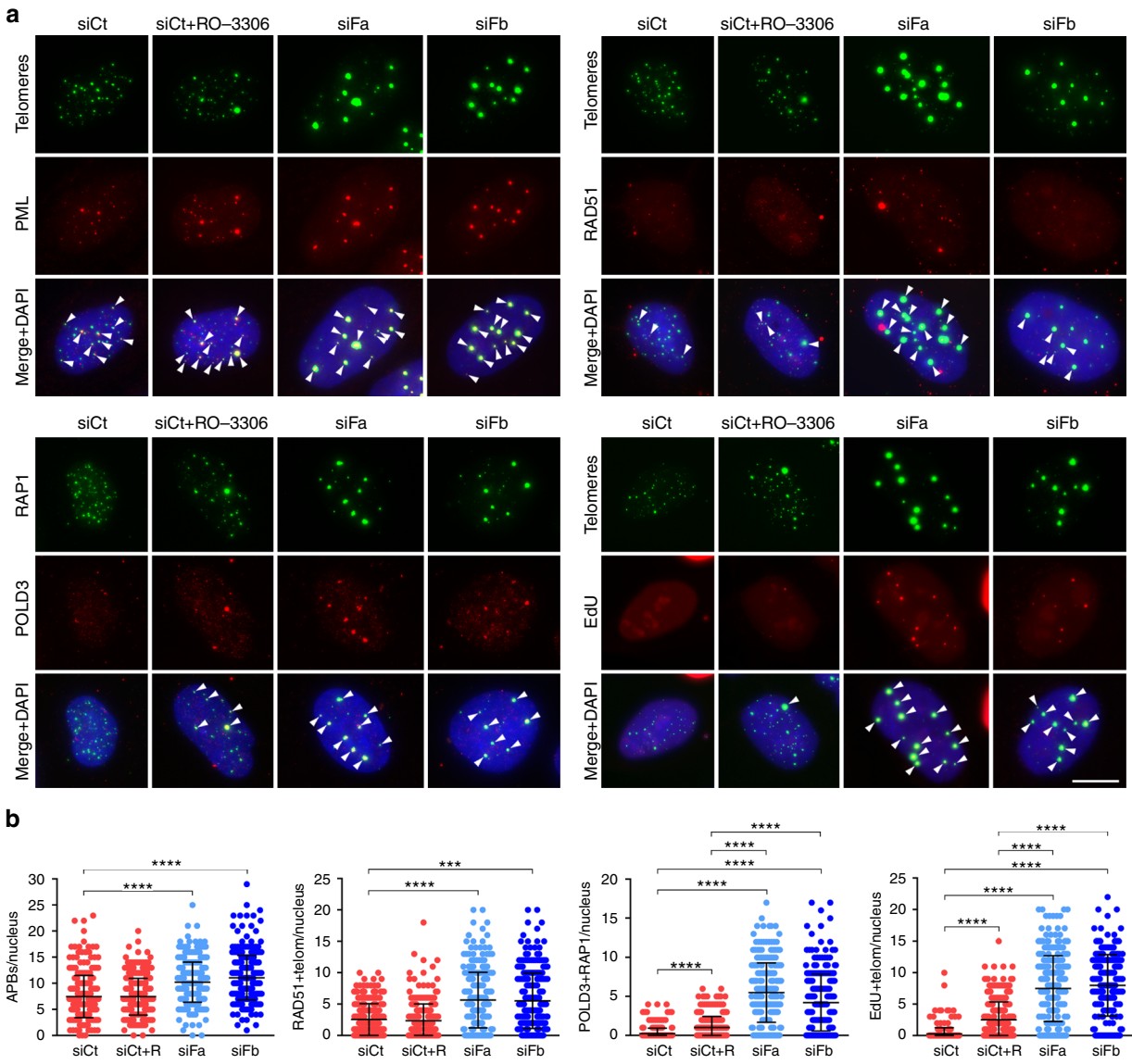

**Fig. 4** FANCM depletion increases ALT features. **a** Upper panels: examples of PML or RAD51 immunostaining (red) combined with telomeric DNA FISH (green); lower left panel: examples of POLD3 immunostaining (red) combined with RAP1 immunostaining (green); lower right panel: examples of EdU detection (red) combined with and telomeric DNA FISH (green). Experiments were performed on U2OS cells transfected with the indicated siRNAs and harvested 48 h after transfection. SiCt-transfected cells treated with RO-3306 were included. In the merge panel, DAPI-stained DNA is also shown (blue). Arrowheads point to colocalization events. Scale bar: 10 µm. **b** Quantifications experiments as in (**a**). Each dot represents an individual nucleus. A total of at least 300 nuclei from three independent experiments were analyzed for each sample. Bars and error bars are means and SDs. P values were calculated with a Mann−Whitney U test. ***$P < 0.001$, ****$P < 0.0001$. Source data are provided as a Source Data file

chromosome ends (Supplementary Fig. 5A and B). C-rich ssDNA-containing ECTRs and DNA threads may explain the well-retained DNA molecules observed in our TRF analysis and the increased telomeric ssDNA observed in our dot-blot analysis (Fig. 5a, b). We did not observe an increase in the incidence of TFEs in FANCM-depleted U2OS cells (Supplementary Fig. 5A and B).

**FANCM regulates BLM in ALT cells.** FANCM and BLM were reported to collaborate in maintaining ALT telomeres[17]. We depleted FANCM in U2OS and HeLa cells and performed indirect IF using anti-BLM and anti-TRF2 antibodies. Because FANCM is necessary for BLM recruitment to damage sites induced by stalled replication[34], we included cells treated with the topoisomerase I inhibitor Camptothecin (CPT; Supplementary Fig. 6A). CPT induced robust formation of nuclear

(nontelomeric) BLM foci in siCt-transfected U2OS and HeLa cells, but not in siFa-transfected cells (Supplementary Fig. 6B and C). On the other hand, BLM TIFs were already abundant in siCt-transfected U2OS and only rarely observed in siCt-transfected HeLa cells, and FANCM depletion increased the number of BLM TIFs in U2OS cells (Supplementary Fig. 6B and C). CPT treatment marginally affected TIF frequencies in all samples (Supplementary Fig. 6B and C). BLM nuclear relocalization occurred without major changes in total protein levels (Supplementary Fig. 6A). Hence, we confirm that FANCM depletion causes BLM accumulation at ALT telomeres[17], while it prevents it at non-telomeric sites of damage both in ALT and Tel+ cells[34]. The telomeric accumulation of BLM upon FANCM depletion might involve reported interactions with TRF1 and TRF2[47].

We then depleted FANCM and BLM simultaneously in U2OS cells (Fig. 6a). BLM depletion alone did not alter cell-cycle

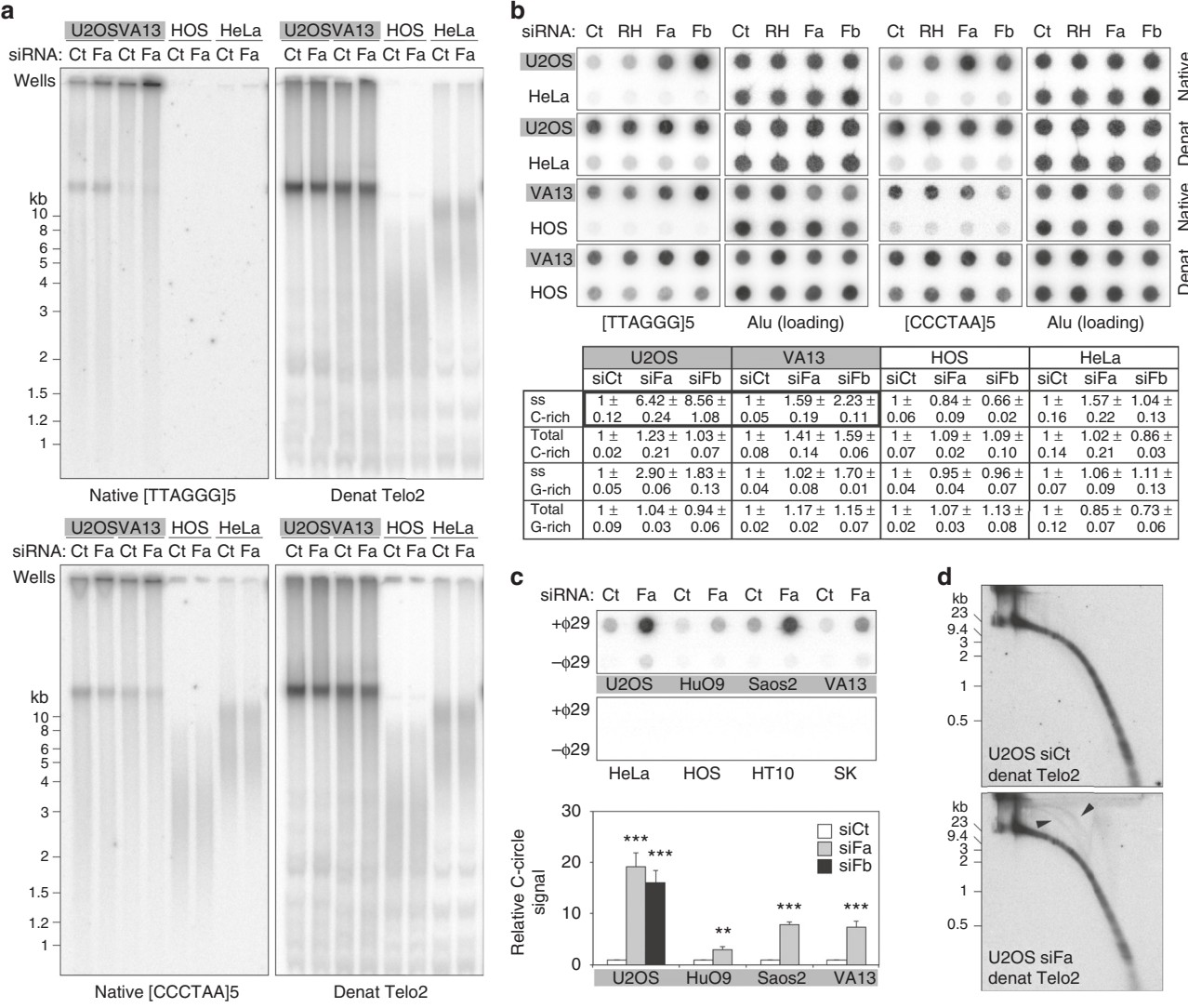

**Fig. 5** FANCM restricts telomeric ssDNA and ECTRs in ALT cells. **a** TRF analysis of the indicated ALT (gray backgrounds) and Tel+ siRNA-transfected cells. Genomic DNA was prepared 48 h after transfection, restriction digested and hybridized in-gel in native conditions to radiolabeled oligonucleotides comprising five telomeric G-rich or C-rich repeats ([TTAGGG]5 and [CCCTAA]5, respectively). After signal acquisition, gels were denatured and rehybridized to a long radiolabeled telomeric probe (Telo2). The position of the wells and the sizes in kb of a molecular weight marker are indicated on the left of the gels. **b** Dot-blot hybridizations of digested genomic DNA from cells as in (**a**). Control transfections with siRNAs against RNaseH1 (siRH) were also included. Native or denatured DNA was first hybridized to radiolabeled telomeric oligonucleotides. After signal acquisition, membranes were denatured and rehybridized to radiolabeled Alu repeat oligonucleotides (loading). For quantifications (table below), telomeric signals were normalized through the corresponding Alu signal and expressed relative to siCt-transfected samples. Means and SDs from three technical replicates are indicated. Note the accumulation of C-rich ssDNA in FANCM-depleted ALT cells (thick borders). **c** C-circle assay analysis of genomic DNA from the indicated siRNA-transfected cells harvested 48 h after transfection. Products were dot-blotted and hybridized to a radiolabeled Telo2 probe. Control reactions were performed without phi29 polymerase (Φ29). Note that Tel+ cells had no detectable signals. The graph at the bottom shows quantifications of C-circle signals relative to siCt-samples. Bars and error bars are means and SDs from three independent experiments. P values were calculated with a two-tailed Student's t test. **P < 0.005, ***P < 0.001. **d** 2D gel electrophoresis of genomic DNA from siRNA-transfected U2OS cells as in **a**. DNA was denatured and hybridized to a radiolabeled Telo2 probe. Arrowheads point to arches corresponding to circular DNA. Source data are provided as a Source Data file

distribution and number of colonies formed, while it decreased proliferation rates and only minimally augmented the fraction of PI-permeable cells (Fig. 6b–d; Supplementary Figs. 1C, 2A). Unexpectedly, FANCM and BLM codepletion resulted in a partial rescue of the aberrant cell-cycle distribution and cell proliferation and viability deriving from depleting FANCM (Fig. 6b–d; Supplementary Figs. 1C, 2A). Moreover, BLM depletion halved the incidence of pS33 TIFs in cells depleted for FANCM (Fig. 6e, f). These results establish that BLM depletion alleviates the adverse effects exerted by FANCM deficiency on ALT cells.

**FANCM suppresses TERRA and telR-loops in ALT cells.** To test whether FANCM suppresses telomere replication stress in ALT cells by regulating TERRA and/or telR-loops, we first performed TERRA northern blot and found that the levels of this lncRNA were 3.5 and 2.5 folds higher in siFa- and siFb-transfected cells, respectively, than in siCt-transfected ones. TERRA species up to ~2 kb in length were the most affected (Fig. 7a). We then performed in vitro R-loop resolution assays using telR-loop-containing plasmids generated by T7 transcription of a telomeric tract of approximately 1 kb[11]. We used two

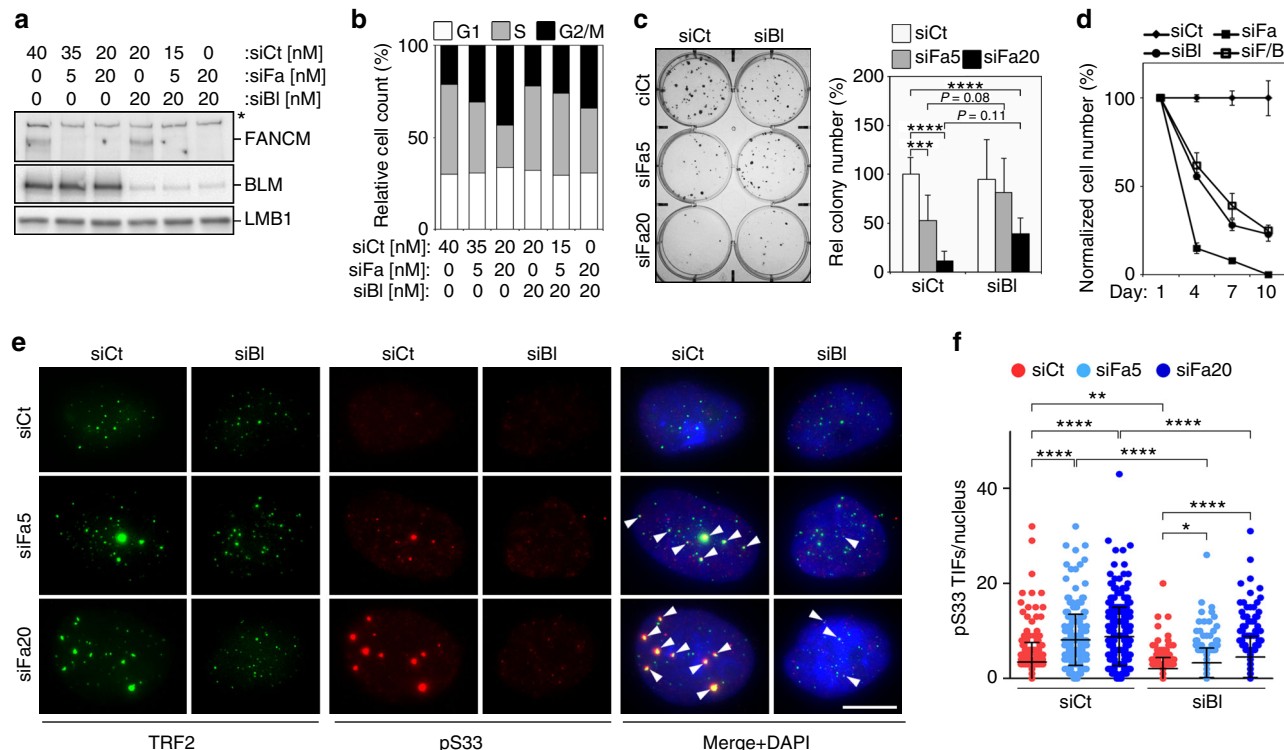

**Fig. 6** BLM depletion substantially averts the phenotypes associated with FANCM depletion. **a** Western blot analysis of FANCM and BLM in U2OS cells transfected with siFa, anti-BLM siRNAs (siBl), and siCt. Two different concentrations (5 and 20 nM) of siFa were used. Cells were harvested 48 h after transfection. LMB1 serves as loading control. The asterisk indicates a band cross-reacting with the anti-FANCM antibody. **b** Quantifications of FACS profiles of cells as in (**a**) stained with PI. The graph shows the percentage of cells in G1, S and G2/M phases from one representative experiment. **c** Example of colony formation assays using cells as in (**a**). siFa5: 5 nM siRNA, siFa20: 20 nM siRNA. The graph on the right shows colony numbers relative to siCt-transfected samples. Bars and error bars are means and SDs from four independent experiments. P values were calculated with a two-way ANOVA followed by Tukey's HSD. **d** Growth curves of U2OS cells transfected with the indicated siRNAs (20 nM each) every 3 days. Cell numbers are expressed relative to siCt-transfected cells. Data points and error bars are means and SDs from three independent experiments. SiCt and siFa curves are the same as the ones shown in Fig. 1f. **e** Examples of pS33 immunostaining (red) combined with TRF2 immunostaining (green) on cells as in (**a**). In the merge panel, DAPI-stained DNA is also shown (blue). Arrowheads point to pS33 TIFs. Scale bar: 10 μm. **f** Quantifications of numbers of pS33 TIFs per nucleus in cells as in (**a**). Each dot represents an individual nucleus. A total of at least 300 nuclei from three independent experiments were analyzed for each sample. Bars and error bars are means and SDs. P values were calculated with a two-way ANOVA followed by Tukey's HSD. *P < 0.05, **P < 0.005, ***P < 0.001, ****P < 0.0001. Source data are provided as a Source Data file

plasmids with different insert orientations as to produce transcripts containing TERRA-like, G-rich RNA repeats, or complementary C-rich transcripts (Fig. 7b). As expected[11], G-rich transcripts were less efficiently produced than C-rich ones (Fig. 7b). TelR-loop plasmids were incubated with recombinant FANCM in heterodimer with its stabilization partner FAAP24, with or without ATP and then resolved in agarose gels. FANCM promoted complete release of both G-rich and C-rich transcripts from R-loop-plasmids without RNA degradation and in an ATP-dependent manner (Fig. 7b). Thus, FANCM efficiently unwinds the RNA moiety of telR-loops in vitro.

To examine telR-loops in FANCM-depleted U2OS cells, we performed DNA:RNA immunoprecipitations (DRIP) using the monoclonal antibody S9.6[48]. Dot-blot hybridization detected telomeric DNA in immunoprecipitated material from all samples, with a ~3-fold increase in siFa and siFb samples (Fig. 7c). Treatment of nucleic acids with recombinant RNaseH prior to antibody incubation largely abolished hybridization signals, confirming that they emanated from DNA:RNA hybrids (Fig. 7c). We also performed native DNA FISH using G-rich telomeric probes on interphase nuclei treated or not with RNaseH[11]. A punctate staining corresponding to C-rich telomeric DNA was already visible in untreated siCt-transfected cells, and its intensity was higher in RNaseH-treated cells likely due to degradation of

TERRA transcripts within telR-loops and consequent increased binding sites for the probe (Fig. 7d, e). In untreated siFa-transfected cells, the C-rich ssDNA signal was more prominent than in control cells and it was further augmented by RNaseH treatment (Fig. 7d, e). The total number of foci per cell was higher in FANCM-depleted cells but was not affected by RNaseH treatment (Fig. 7d, e). We conclude that FANCM suppresses TERRA and TERRA-containing telR-loops in ALT cells. Considering the ability of FANCM to resolve telR-loops in vitro (Fig. 7b) and the localization of FANCM to telomeres (Fig. 2c, d), we propose that FANCM directly resolves telR-loops on telomeric chromatin. The more prominent C-rich ssDNA signal already present in FANCM-depleted cells not treated with RNaseH (Fig. 7d, e) might originate from gaps in DNA replication or cellular degradation of the RNA moiety of telR-loops. Also, although we refer to the telomeric RNA:DNA hybrid structures arising upon FANCM depletion as telR-loops, our experiments do not distinguish between conventional R-loops, three-stranded nucleic acids comprising an RNA:DNA hybrid and a displaced ssDNA, and ds RNA:DNA hybrids devoid of a displacement loop.

**FANCM averts telR-loop-induced telomeric replication stress.** We speculated that FANCM suppresses telomeric replication

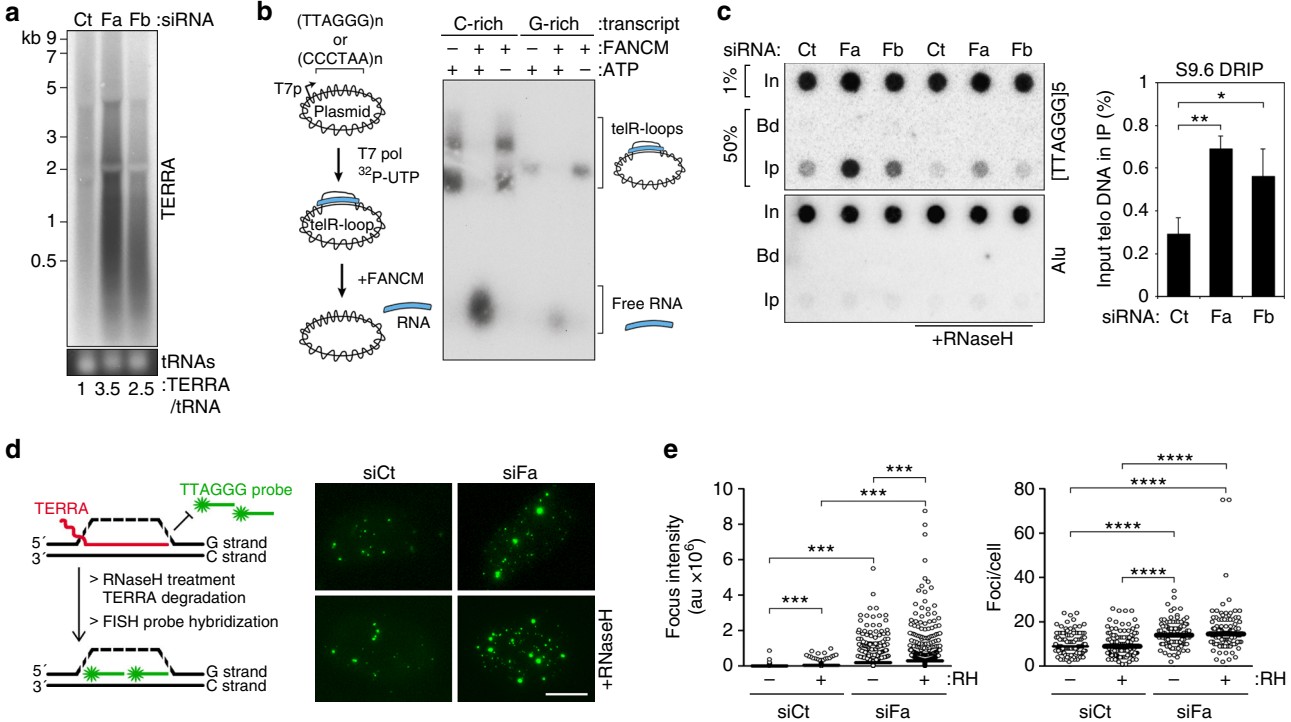

**Fig. 7** FANCM restricts TERRA and telR-loops in ALT cells. **a** TERRA northern blot using RNA from U2OS cells harvested 48 h after siRNA transfection. Sizes of a molecular weight marker are on the left. Numbers at the bottom are quantifications of TERRA signals normalized through the corresponding tRNA signals and expressed relative to the siCt sample. **b** Schematic representation of how telR-loops were generated and unwound in vitro. The gel on the right is an example of a telR-loop unwinding assay performed with telR-loop plasmids (1 nM) and purified recombinant FANCM-FAAP24 (2.5 nM). Note that FANCM unwinds telR-loops only when ATP is included in the reaction. **c** Dot-blot hybridization of DRIPs in U2OS cells as in (**a**) using radiolabeled G-rich telomeric oligonucleotides. After signal acquisition, membranes were denatured and rehybridized to radiolabeled Alu repeat oligonucleotides. In Input, Bd only beads control, Ip S9.6 immunoprecipitation. Signals are graphed on the right as the fraction of In found in the corresponding Ip samples, after subtraction of Bd-associated signals. Alu signals are not included in the quantification. Bars and error bars are means and SDs from four independent experiments. *P* values were calculated with a two-tailed Student's *t* test. **d** Schematic representation of the protocol for native FISH. The displaced DNA strand is indicated by a dotted line because the same protocol allows detection also of C-rich DNA engaged in RNA:DNA hybrids devoid of a displacement loop. Images on the right are examples of native FISH on siRNA-transfected U2OS cells as in (**a**). Signals from the G-rich telomeric probe and therefore deriving from C-rich ssDNA are in green. Scale bar: 10 μm. **e** Quantifications of experiments as in (**d**). 3D images were sum projected and integrated intensities of FISH signal were measured within individual nuclei identified by DAPI staining (not shown) and background subtracted. Each dot represents an individual nucleus. A total of 100–120 nuclei were analyzed for each sample. One representative experiment is shown. Bars are means. *P* values were calculated with a Mann–Whitney *U* test. \**P* < 0.05, \*\**P* < 0.005, \*\*\**P* < 0.001, \*\*\*\**P* < 0.0001. Source data are provided as a Source Data file

stress by dismantling telR-loops. We depleted FANCM in U2OS cells infected with retroviruses expressing an siRNA-resistant, V5 epitope-tagged FANCM variant (V5-FANCM WT) or an ATPase/translocase inactive counterpart unable to resolve R-loops (V5-FANCM K117R[36]). Both variants were expressed at higher levels than endogenous FANCM (Fig. 8a). Confirming the specificity of our siRNAs, V5-FANCM WT largely averted G2/M arrest and accumulation of ps33 TIFs and APBs in siFa-transfected cells (Fig. 8b, c; Supplementary Figs. 1D, 7A). On the contrary, cell-cycle distribution and incidence of pS33 TIFs and APBs were similar in V5-FANCM K117R and ev control cells transfected with siFa (Fig. 8b, c; Supplementary Figs. 1D, 7A).

We then exploited the ability of overexpressed RNaseH1 to suppress telR-loops in cells[11,39,49]. We depleted FANCM alone or in combination with BLM in U2OS cells infected with retroviruses driving overexpression of MYC epitope-tagged RNaseH1 (MYC-RH1 WT) or a catalytically dead counterpart (MYC-RH1 D145A), or with ev control retroviruses (Fig. 8d). MYC-RH1 WT further enhanced the rescue of G2/M arrest defect in cells codepleted for FANCM and BLM (Fig. 8e; Supplementary Fig. 1E). Decreased frequencies of FANCM depletion-induced pS33 TIFs were measured in cells expressing MYC-RH1 WT as compared to ev-infected cells (Fig. 8f; Supplementary Fig. 7B). In

cells codepleted for FANCM and BLM and overexpressing MYC-RH1 WT, pS33 TIFs were restored to levels similar to ev control cells (Fig. 8f; Supplementary Fig. 7B). In all experiments, MYC-RH1 D145A failed to function as its catalytically active counterpart (Fig. 8e, f; Supplementary Figs. 1E, 7B). These results indicate that the telomeric replication stress arising upon FANCM depletion is suppressed by FANCM enzymatic activity and stems from unresolved telR-loops and uncontrolled BLM.

## Discussion

We demonstrate here that, in absence of FANCM, ALT cells experience severe telomeric replication stress and activate an ATR-mediated DNA damage signaling, which is likely the trigger of the observed G2/M arrest and cell death[50]. The fact that overexpression of TRF1 renders ALT cells less sensitive to FANCM depletion (Fig. 1g, h) and the lack of accumulation of nontelomeric pSer33 and 53BP1 foci in FANCM-depleted ALT cells (Fig. 2a) confirm the centrality of the signal emanating from damaged telomeres in promoting G2/M arrest. However, FANCM may serve essential functions in ALT cells also outside telomeres, as suggested by its physical interaction with Alu repeat DNA.

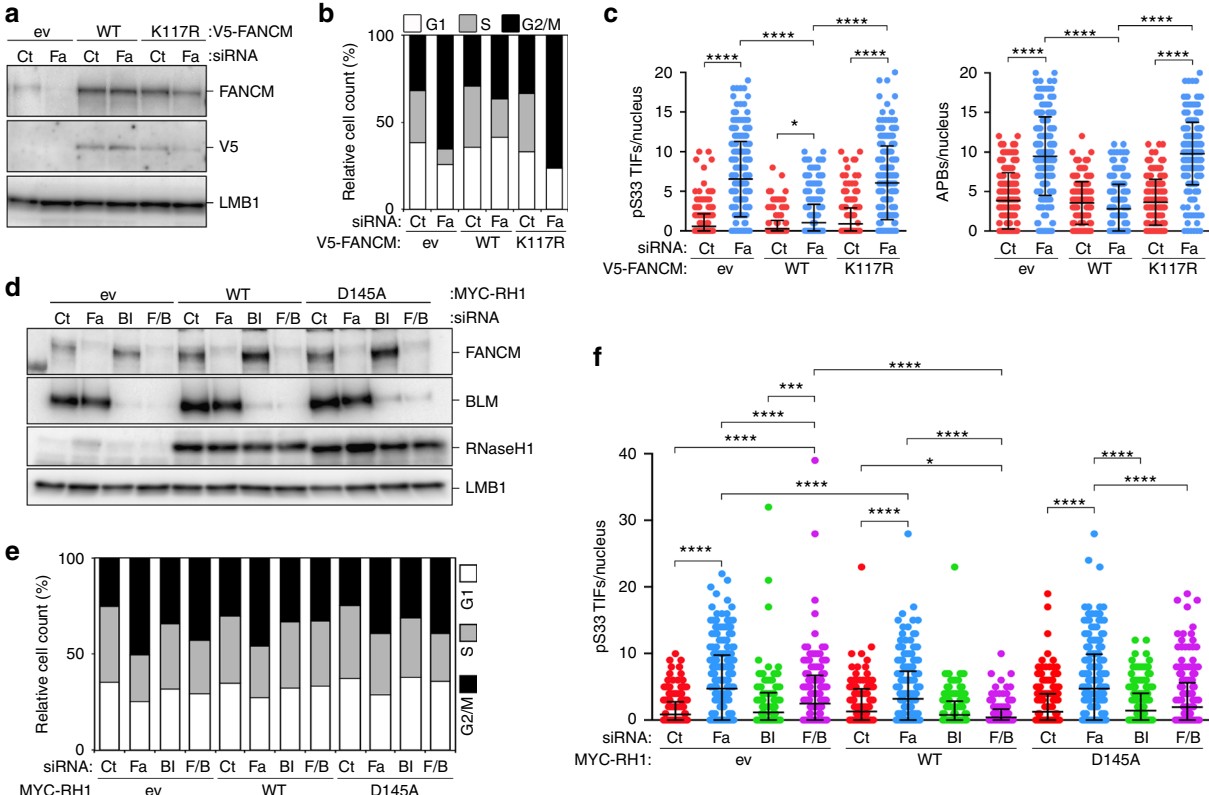

**Fig. 8** Deregulated telR-loops contribute to the replication stress arising upon FANCM depletion in ALT cells. **a** Western blot analysis of U2OS cells infected with retroviruses expressing V5 epitope-tagged FANCM variants or empty vector (ev) control retroviruses. WT wild type, K117R ATPase/translocase dead FANCM. Five days after infections cells were transfected with the indicated siRNAs and harvested 48 h later. LMB1 serves as loading control. **b** Quantifications of FACS profiles of cells as in (**a**) stained with PI. The graph shows the percentage of cells in G1, S and G2/M phases from one representative experiment. **c** Quantifications of numbers of pS33 TIFs and APBs per nucleus in cells as in (**a**). Each dot represents an individual nucleus. A total of at least 300 nuclei from three independent experiments were analyzed for each sample. Bars and error bars are means and SDs. *P* values were calculated with a two-way ANOVA followed by Tukey's HSD. **d** Western blot analysis of U2OS cells infected with retroviruses expressing MYC epitope-tagged RNaseH1 (RH1) variants or ev control retroviruses. D145A: endoribonuclease dead RNaseH1. Five days after infections cells were transfected with the indicated siRNAs and harvested 48 h later. siF/B: combined siFa and siBl. LMB1 serves as loading control. **e** Quantifications of FACS profiles of cells as in (**d**) stained with PI. The graph shows the percentage of cells in G1, S and G2/M phases from one representative experiment. **f** Quantifications of numbers of pS33 TIFs per nucleus in cells as in (**d**). Each dot represents an individual nucleus. A total of at least 300 nuclei from three independent experiments were analyzed for each sample. Bars and error bars are means and SDs. *P* values were calculated with a two-way ANOVA followed by Tukey's HSD. Comparisons between D145A and ev and WT are not indicated. *$P < 0.05$, ***$P < 0.001$, ****$P < 0.0001$. Source data are provided as a Source Data file

We also confirm that FANCM is not essential in all cells, as proliferation and viability of Tel+ cells was not majorly affected by FANCM depletion. Consistently, adult humans carrying biallelic loss of function FANCM mutations were reported[51,52]. Moreover, Tel+ human colorectal carcinoma cells, mouse embryonic fibroblasts and chicken lymphoblasts knocked-out for FANCM were successfully generated and proliferated normally unless challenged with DNA damage[53–55]. As such, FANCM represents an attractive target in ALT cancer therapy. While it is true that FANCM deficiency is associated with higher risk of breast and liver cancer[51,55], the irreversible lesions rapidly inflicted by FANCM depletion on ALT cells indicate that short-term inhibition of FANCM may efficiently eradicate ALT tumors in absence of secondary effects. On the other side, the previously proposed therapy based on coinhibition of FANCM and BLM should be avoided[17].

We also show that FANCM suppresses ALT-associated features, including clustering of telomeres in PML-, POLD3- and RAD51-containing APBs, and production of ECTRs which include C-circles but possibly other forms (Figs. 4a, b, 5c, d). Because the same features were not evident when we depleted FANCM in Tel+ cells, FANCM deficiency alone is not sufficient

to initiate ALT de novo. Further in line with FANCM limiting ALT, its depletion increased synthesis of telomeric DNA outside of S phase (Fig. 4a, b) and led to appearance of DNA threads (Supplementary Fig. 5A and B), which may represent intermediates of intermolecular recombination events. We propose that FANCM suppresses POLD3-dependent telomeric BIR in G2 and possibly MiDAS[24–26]. Consistently, deletion of the yeast helicase Mph1, the *Saccharomyces cerevisiae* FANCM ortholog[56], directed repair of an HO endonuclease-induced DSB towards BIR[57]. Moreover, Mph1 overexpression inhibited BIR at intra-chromosomal DSBs[58], while it did not prevent insurgence of telomerase-deficient type II survivors, which are ALT yeasts that maintain their telomeres though BIR[58–62]. Finally, MphI localizes to short telomeres in an R-loop-dependent manner[63]. FANCM proteins appear to play specific roles at ALT and uncapped telomeres that are different from the ones that they exert at intra-chromosomal damage sites, and that are mediated by R-loops. TelR-loops may directly promote recruitment and/or stabilization and in turn activation of FANCM at telomeres in human ALT cells, thus regulating POLD3-dependent telomeric BIR. The relevance of RAD51 accumulation in APBs when FANCM is depleted remains unclear (Fig. 4a, b). RAD51 could mediate the

telomere clustering observed in FANCM-depleted cells or other molecular events which we did not investigate, such as sister telomere exchanges.

Despite increasing ALT activity, FANCM depletion did not elicit major gain of total (G-rich plus C-rich) telomeric DNA (Fig. 5b). It is possible that in our experiments the amount of newly produced telomeric DNA was below the detection limit. Additionally, de novo synthesis of telomeric DNA in FANCM-depleted cells is likely counterbalanced by incomplete semi-conservative replication of telomeric DNA in S phase[17] and by elimination of ECTRs from cells. The exact mechanism by which circular ECTRs are generated remains to be elucidated; nevertheless, they are associated with features that increase in FANCM-depleted ALT cells, including replication stress, activated ATR, C-rich telomeric ssDNA and telR-loops[11,64–66]. One observation of our study is that FANCM depletion does not alter TFE frequencies (Supplementary Fig. 5A and B). This suggests that the observed ECTRs do not derive from excision of entire telomeric tracts.

The replication stress arising at ALT telomeres upon FANCM depletion mainly originates from two sources, deregulated BLM and telR-loops. Given the increased BLM recruitment to ALT telomeres when FANCM is depleted (Supplementary Fig. 6A–C), FANCM could directly displace BLM from telomeres. Alternatively, FANCM could suppress the triggers provoking BLM recruitment such as arrested telomeric replication forks or R- and D-loop intermediates. BLM activity promotes ALT by supporting telomeric recombination and BIR-based synthesis of telomeric DNA possibly by resolving recombination intermediates formed during BIR-associated strand invasion as part of the BLM-TOP3A-RMI (BTR) dissolvase complex[15,67,68]. Consistently, FANCM depletion increases telomere synthesis outside of S-phase (Fig. 4a, b). Moreover, because BLM mediates long-range resection of DNA ends[69,70], hyperactive BLM is likely to directly contribute to the production of telomeric ssDNA when FANCM is depleted. This is consistent with the low levels of pS33 TIFs detected in cells double depleted for FANCM and BLM (Fig. 8f). As FANCD2 has also been shown to suppress BLM toxicity in ALT cells[68], one could speculate that this may be a general role for the FA pathway. Nevertheless, the ATPase/translocase activity of FANCM is dispensable for FANCD2 monoubiquitination[31], and overexpression in U2OS cells of a variant of FANCM unable to recruit the FA complex to chromatin did not suppress any ALT-associated feature (accompanying manuscript, "The FANCM-BLM-TOP3A-RMI complex suppresses alternative lengthening of telomeres (ALT)", Lu et al.). It seems therefore unlikely that the entire FA complex functions to maintain ALT.

As for the nature of telR-loops in FANCM-depleted cells, our data suggest that they accumulate because they are not properly dismantled at telomeric chromatin by the ATPase/translocase activity of FANCM (Fig. 7b–e). TelR-loops could be generated cotranscriptionally, as telomeric DNA is a difficult substrate for RNA polymerases[11,49]. The increased short TERRA species observed in FANCM-depleted ALT cells (Fig. 7a) might indeed derive from premature termination of telomere transcription due to improper telR-loop resolution. Also, the accumulation of pS33 indicates that FANCM most likely resolves telR-loops during S-phase. Thus, improper telR-loop resolution can at least in part explain the diminished efficiency of replication fork progression through the telomeric tract in FANCM-depleted cells[17]. Moreover, some FANCM-depletion-associated features of replication-stress, accumulation of C-rich telomeric ssDNA and C-circles (Fig. 5a–e), are more evident in U2OS cells than in other ALT-cells. This might be explained by the fact that TERRA and telR-loops are particularly abundant in U2OS cells[11].

Due to the fast and dramatic response of ALT cells to FANCM inactivation (Fig. 1b–f), our studies had to be performed using siRNAs rather than CRISPR/Cas9-based gene inactivation, obfuscating the analysis of genetic interactions. Nevertheless, the synergism between BLM depletion and RNaseH1 overexpression in suppressing replicative stress in FANCM-depleted cells suggests that BLM activity and telR-loops may be functionally related. Although BLM suppresses R-loops genome-wide[71], it could promote telR-loop formation specifically in ALT cells, for example by generating C-rich ssDNA followed by TERRA annealing. Conversely, telR-loops could recruit BLM to telomeres by stalling telomeric replication forks or by forming D-loop mimicking structures. Future studies, possibly utilizing conditional knockout cells for FANCM and BLM, should refine this intriguing cellular scenario and open the way to novel avenues for curing ALT cancers.

## Methods

**Cell lines and culture conditions**. HeLa cervical cancer, HT1080 fibrosarcoma, and HEK293 embryonic kidney cells were purchased from ATCC. U2OS osteosarcoma cells were a kind gift from M. Lopes (IMCR, Zurich, Switzerland). HuO9, Saos2 and HOS osteosarcoma cells were a kind gift from B. Fuchs (Balgrist University Hospital, Zurich, Switzerland). WI-38 VA13 in vitro SV40-transformed lung fibroblasts were a kind gift from A. Londoño-Vallejo (CNRS, Paris, France). SKNAS neuroblastoma cells were a kind gift from O. Shakhova (University Hospital Zurich, Switzerland). HeLa, HT1080, HEK293, U2OS, and WI-38 VA13 cells were cultured in high glucose DMEM, GlutaMAX (Thermo Fisher Scientific) supplemented with 10% tetracycline-free fetal bovine serum (Pan BioTech) and 100 U/ml penicillin-streptomycin (Thermo Fisher Scientific). HuO9, Saos2, HOS and SKNAS cells were cultured in high glucose DMEM/F12, GlutaMAX (Thermo Fisher Scientific), supplemented with 10% tetracycline-free fetal bovine serum (Pan BioTech), 100 U/ml penicillin-streptomycin (Thermo Fisher Scientific) and non-essential amino acids (Thermo Fisher Scientific). Mycoplasma contaminations were tested using the VenorGeM Mycoplasma PCR Detection Kit (Minerva Biolabs) according to the manufacturer's instructions. When indicated, cells were incubated with 1 µM camptothecin (Sigma-Aldrich) for 3 h, 0.2 mM hydroxyurea (Sigma-Aldrich) for 16 h, 20 µM BIBR 1532 (Merck Millipore) for 7 days, and 10 µM RO-3306 (Selleckchem) for 18 h.

**Ectopic protein expression**. For FANCM complementation experiments, siFa- and siFb-resistant cDNAs coding for N-terminally V5-tagged FANCM variants were synthesized at GenScript and cloned into the into the lentiviral vector pLVX-TetOne-Puro (Clontech). The obtained plasmids, pLVX-V5FANCM and pLVX-V5FANCMK117R, were used to produce lentiviruses and infect U2OS cells, followed by selection in medium containing 1 µg/ml puromycin (Merck Millipore). Experiments were performed in medium containing 1 µg/ml doxycycline (Sigma-Aldrich). For RNaseH1 overexpression, U2OS cells were infected with retroviruses produced using the pLHCX-MYC-RH1WT and pLHCX-MYC-RH1D145A plasmids[11], followed by selection in medium containing 200 µg/ml hygromycin B (VWR). For TRF1 overexpression, U2OS cells were infected with retroviruses produced using the pLPC-NFLAG-TRF1 (a kind gift from T. de Lange, Addgene plasmid # 16058), followed by puromycin selection. Transgene expression was validated by western blotting. For telomerase overexpression, U2OS and HeLa cells were infected with retroviruses produced using the pBABEpuroUThTERT + U3-hTR-500 plasmid (a kind gift from K. Collins, Addgene plasmid #27665), followed by puromycin selection. Viruses were produced in HEK293 cells according to standard procedures. FANCM, RNaseH1 and TRF1 ectopic expression was validated by western blotting (see below). hTERT and hTR expression was validated by quantitative RT-PCR on total RNA using the following oligonucleotides: hTERT-for, 5′-agagtgtctggagcaagttgc-3′; hTERTrev, 5′-cgtagtccatgttcacaatcg-3′; hTRfor, 5′-gtggtggccatttttgtctaac-3′; hTRrev, 5′-tgctctagaatgaacggtggaa-3′; ActinB1for, 5′-tccctggagaagagctacga-3′; Actin B1rev, 5′-agcactgtgttggcgtacag-3′. Actin B1 was used as a normalizer.

**siRNA-mediated protein depletion**. DsiRNAs (Integrated DNA Technologies) were transfected using the Lipofectamine RNAiMAX reagent (Invitrogen) according to the manufacturer instructions. DsiRNAs were used at a final concentration of 20 nM unless otherwise indicated. Medium was changed 5 h after transfection and samples collected 48 h after transfection unless otherwise indicated. The following mRNA target sequences were used:

siFa: 5′- GGATGTTTAGGAGAACAAAGAGCTA-3′;
siFb: 5′-CCCATCAAATGAAGATATGCAGAAT-3′;
siBl: 5′-GCTAGGAGTCTGCGTGCGAGGATTA-3′;
siATRXa: 5′-GAGGAAACCUUCAAUUGUAACAAAGUA-3′;
siATRXb: 5′-UGCAAGCUCUAUCAGUACUACUUAGAU-3′;
siTRF1: 5′-CUUUCUUUCUUAUUUAAGGUCUUGUUGC-3′;

siRNaseH1: 5′-UUGUCUAAUGCCUACAUUUAAAGGAUG-3′.
NC1 negative control (51-01-14-03) was used as siCt.

**Cell proliferation and viability assays**. For colony-forming assays, cells were transfected with siRNAs and 24 h later 300−500 cells were plated in 3 cm dishes and grown until visible colonies were formed. Cells were stained in 1% Crystal violet, 1% formaldehyde, 1% MeOH (Sigma-Aldrich) for 20 min at room temperature, followed by washes in tap water. Plates were air-dried and photographed with a FluorChem HD2 imaging apparatus (Alpha Innotech). Colonies were counted using ImageJ software. For growth curves, cells were transfected with siRNAs and 24 h later $1 \times 10^5$ cells were seeded in 6 cm dishes and passaged for 10 days. Cells were retransfected with siRNAs and counted every 3 days. For fluorescence-activated cell sorting, cells were trypsinized and pelleted by centrifugation at $500 \times g$ at 4 °C for 5 min. Cell pellets were either left untreated for viability assays or fixed in 70% ethanol at −20 °C for 30 min and treated with 25 μg/ml RNaseA (Sigma-Aldrich) in 1× PBS at 37 °C for 20 min. Cells were then washed in 1× PBS and stained with 20 μg/ml propidium iodide (Sigma-Aldrich) in 1× PBS at 4 °C for 10 min. Flow cytometry was performed on a BD FACSCalibur or a BD Accuri C6 (BD Biosciences). Data were analyzed using FlowJo software.

**Western blotting**. Cells were trypsinized and pelleted by centrifugation at $500 \times g$ at 4 °C for 5 min. Pellets were resuspended in 2× lysis buffer (4% SDS, 20% Glycerol, 120 mM Tris-HCl pH 6.8), boiled at 95 °C for 5 min and centrifuged at $1600 \times g$ at 4 °C for 10 min. Supernatant were recovered and protein concentrations were determined by Lowry assay using bovine serum albumin (BSA; Sigma-Aldrich) as standard. 20–40 μg of proteins were supplemented with 0.004% Bromophenol blue and 1% β-Mercaptoethanol (Sigma-Aldrich), incubated at 95 °C for 5 min, separated in 6 or 10% polyacrylamide gels, and transferred to nitrocellulose membranes (Maine Manufacturing, LLC) using a Trans-Blot SD Semi-Dry Transfer Cell apparatus (Bio-Rad). The following primary antibodies were also used: mouse monoclonal anti-FANCM (CV5.1[72], 1:1000 dilution); mouse monoclonal anti-Golgin 97 (Molecular Probes, A-21270, 1:5000 dilution); rabbit polyclonal anti-KAP1 (Bethyl Laboratories, A300-274A, 1:2000 dilution); rabbit polyclonal anti-pKAP1 (Ser 824) (Bethyl Laboratories, A300-767A, 1:2000 dilution); mouse monoclonal anti-CHK1 (Santa Cruz Biotechnology, sc-8408, 1:1000 dilution); rabbit monoclonal anti-pCHK1 Ser 345 (Cell Signaling, 2348, 1:500 dilution); rabbit polyclonal anti-RPA32 (Bethyl Laboratories, A300-244A, 1:3000 dilution); rabbit polyclonal anti-pRPA32 Ser 33 (Bethyl Laboratories, A300-246A, 1:1000 dilution); rabbit polyclonal anti-Lamin B1 (GeneTex, GTX103292, 1:1000 dilution); rabbit polyclonal anti-RNaseH1 (GeneTex, GTX117624, 1:500 dilution); mouse monoclonal anti-beta Actin (Abcam, ab8224, 1:5000 dilution); rabbit polyclonal anti-BLM (Bethyl Laboratories, A300-110A, 1:3000 dilution); rabbit polyclonal anti-PML (a kind gift from M. Carmo-Fonseca, iMM, Lisbon, Portugal, 1:1000 dilution); rabbit polyclonal anti-PARP1 (Cell Signaling, 9542, 1:1000 dilution); mouse monoclonal anti-POLD3 (Novus Biologicals, H00010714-M01, 1:500 dilution); rabbit polyclonal anti-ATRX (Bethyl Laboratories, A301-045A-T, 1:1000 dilution); sheep polyclonal anti-TRF1 (R&D Systems, AF5300, 1:1000 dilution). Secondary antibodies were HRP-conjugated goat anti-mouse and goat anti-rabbit IgGs (Bethyl Laboratories, A90-116P and A120-101P, 1:2000 dilution) and HRP-conjugated donkey anti-sheep IgG (Novus Bio, NBP1-75437, 1:3000 dilution). Signal detection was performed using the ECL detection reagents (GE Healthcare) and a FluorChem HD2 imaging apparatus (Alpha Innotech).

**Fluorescence in situ hybridization (FISH)**. Metaphase spreads were prepared by incubating cells with 200 ng/ml Colchicine (Sigma-Aldrich) for 2–6 h, mitotic cells were harvested by shake-off and incubated in 0.075 M KCl at 37 °C for 10 min. Chromosomes were fixed in ice-cold methanol/acetic acid (3:1) and spread on glass slides. Slides were treated with 20 μg/ml RNase A (Sigma-Aldrich), in 1× PBS at 37 °C for 1 h, fixed in 4% formaldehyde (Sigma-Aldrich) in 1× PBS for 2 min, and then treated with 70 μg/ml pepsin (Sigma-Aldrich) in 2 mM glycine, pH 2 (Sigma-Aldrich) at 37 °C for 5 min. Slides were fixed again with 4% formaldehyde in 1× PBS for 2 min, incubated subsequently in 70, 90 and 100% ethanol for 5 min each, and air-dried. A Cy3-conjugated C-rich telomeric PNA probe (TelC-Cy3; 5′-Cy3-OO-CCCTAACCCTAACCCTAA-3′; Panagene) diluted in hybridization solution (10 mM Tris-HCl pH 7.2, 70% formamide, 0.5% blocking solution (Roche)) was applied onto the slides followed by one incubation at 80 °C for 5 min and one at room temperature for 2 h. Slides were washed twice in 10 mM Tris-HCl pH 7.2, 70% formamide, 0.1% BSA and three times in 0.1 M Tris-HCl pH 7.2, 0.15 M NaCl, 0.08% Tween-20 at room temperature for 10 min each. For native FISH experiments on interphase nuclei, cells grown on coverslips were incubated in CSK buffer (100 mM NaCl, 300 mM sucrose, 3 mM MgCl₂, 10 mM PIPES pH 6.8, 0.5% Triton-X) for 7 min on ice. Cells were then fixed in 4% formaldehyde in 1× PBS for 10 min and permeabilized with CSK buffer for 5 min at room temperature. RNaseH treatments were performed by incubating slides with 30 U of RNaseH (Takara) in 1× RNaseH buffer or only with buffer at 37 °C for 2 h. Hybridizations and washes were performed as above but using a TYE 563-conjugated G-rich telomeric LNA probe (TelG-TYE 563; 5′-TYE563-T*AGGGT*AGGGT*AGGG-3′, asterisks indicate LNA nucleotides; Exiqon). DNA was counterstained with 100 ng/ml DAPI (Sigma-Aldrich) in 1× PBS and slides were mounted in Vectashield (Vectorlabs).

Images were acquired with an Olympus IX 81 microscope equipped with a Hamamatsu ORCA-ER camera and a ×60/1.42NA oil PlanApo N objective, or a Zeiss Cell Observer equipped with a cooled Axiocam 506 m camera and a ×63/1.4NA oil DIC M27 PlanApo N objective. Image analysis was performed using ImageJ and Photoshop software.

**Combined FISH and EdU incorporation/detection**. Twenty-four hours after siRNA transfection, cells were incubated in 10 μM RO-3306 (Selleckchem). 21.5 h later 10 μM EdU (Thermo Fisher Scientific) was added to the culture medium, followed by a 2.5 h incubation. Cells were first stained as for DNA FISH using the TelC-Cy3 probe and then washed twice with 1× PBS followed by EdU detection using the Click-iT EdU Alexa Fluor 488 Imaging Kit (Thermo Fisher Scientific) according to the manufacturer's instructions. DNA was counterstained with 100 ng/ml DAPI in 1× PBS and coverslips were mounted on slides in Vectashield. Image acquisition and analysis were as for DNA FISH.

**Indirect immunofluorescence (IF)**. Cells grown on coverslips were incubated in CSK buffer for 7 min on ice. All subsequent treatments were performed at room temperature. Cells were fixed with 4% formaldehyde (Sigma-Aldrich) in 1× PBS for 10 min, permeabilized with CSK buffer for 5 min, and incubated in blocking solution (0.5% BSA, 0.1% Tween-20 in 1× PBS) for 1 h. Coverslips were incubated in blocking solution containing primary antibodies for 1 h, washed three times with 0.1% Tween-20 in 1× PBS for 10 min each, and incubated with secondary antibodies diluted in blocking solution for 50 min. DNA was counterstained with 100 ng/ml DAPI in 1× PBS. For combined IF and DNA FISH, cells were again fixed with 4% formaldehyde in 1× PBS for 10 min, washed three times with 1× PBS, incubated in 10 mM Tris-HCl pH 7.2 for 5 min and then denatured and hybridized with TelC-Cy3 probes as described above. DNA was counterstained with 100 ng/ml DAPI in 0.1 M Tris-HCl pH 7.2, 0.15 M NaCl, 0.08% Tween-20 and coverslips were mounted on slides in Vectashield. The following primary antibodies were used: rabbit polyclonal anti-pRPA32 pSer 33 (Bethyl Laboratories, A300-246A, 1:1000 dilution); rabbit polyclonal anti-53BP1 (Abcam, ab21083, 1:1000 dilution); mouse monoclonal anti-TRF2 (Millipore, 05-521, 1:500 dilution); rabbit polyclonal anti-BLM (Bethyl, A300-110A, 1:5000 dilution); rabbit polyclonal anti-PML (a kind gift from M. Carmo-Fonseca, iMM, Lisbon, Portugal, 1:500 dilution); mouse monoclonal anti-RAD51 (Abcam, ab213, 1:100 dilution); mouse monoclonal anti-POLD3 (Novus Biologicals, H00010714-M01, 1:100 dilution); rabbit polyclonal anti-RAP1 (Bethyl, A300-306A, 1:500 dilution). Secondary antibodies were Alexa Fluor 568-conjugated donkey anti-rabbit IgGs (Thermo Fisher Scientific, A10042) and Alexa Fluor 488-conjugated donkey anti-mouse IgGs (Thermo Fisher Scientific, A21202). Image acquisition and analysis were as for DNA FISH.

**Genomic DNA analysis**. Genomic DNA was isolated by phenol:chloroform extraction and treatment with 40 μg/ml RNaseA, followed by ethanol precipitation. Reconstituted DNA was digested with HinfI and RsaI (New England Biolabs) and again purified by phenol:chloroform extraction. For TRF analysis, 2 μg of digested DNA were separated on 0.6% agarose gels, which were vacuum-dried at 50 °C for 50 min. Gels were hybridized at 50 °C overnight with telomeric oligonucleotide probes (5′-(TTAGGG)₅−3′ or 5′-(CCCTAA)₅−3′), 5′-end labeled with T4 polynucleotide kinase (New England Biolabs) and [γ-³²P]ATP. Post-hybridization washes were twice in 2× SSC, 0.2% SDS for 20 min and once in 0.5× SSC, 0.2% SDS for 30 min at 50 °C. After radioactive signal acquisition, gels were incubated in denaturing solution (1.5 M NaCl, 0.5 M NaOH) at room temperature for 20 min and then hybridized at 55 °C overnight with a double-stranded telomeric probe (Telo2 probe), radioactively labeled using Klenow fragment (New England Biolabs) and [α-³²P]dCTP. Post-hybridization washes were twice in 2× SSC, 0.2% SDS for 20 min and once in 0.2× SSC, 0.2% SDS for 30 min at 55 °C. For dot-blot hybridizations, 1 μg of genomic DNA digested as above was denatured for 5 min at 98 °C or left untreated and dot-blotted on nylon membranes. Membranes were first hybridized with telomeric oligonucleotide probes as above. After radioactive signal acquisition, gels were incubated in denaturing solution as above and then re-hybridized overnight to radiolabeled Alu-repeat oligonucleotides (5′-GTGATCCGCCCGCCTCGGCCTCCCAAAGTG-3′) at 50 °C. Post-hybridization washes were twice in 2× SSC, 0.2% SDS for 20 min and once in 0.5× SSC, 0.2% SDS for 30 min at 50 °C. For C-circle assays, 150−500 ng of digested DNA were incubated with 7.5 U phi29 DNA polymerase (New England Biolabs) in the presence of dATP, dTTP and dGTP (1 mM each) at 30 °C for 8 h, followed by heat-inactivation at 65 °C for 20 min. Amplification products were dot-blotted onto nylon membranes (GE Healthcare) and hybridized to a radiolabeled Telo2 probe as above. For two-dimensional gel electrophoresis 10 μg of digested DNA were separated on 0.6% agarose gels (pulsed field certified agarose; Bio-Rad) at 30 V for 7 h, followed by excision of the lane and separation of the DNA in the second dimension on 1.1% agarose gels (UltraPure agarose; Life Technologies) at 100 V for 3 h. DNA was then transferred onto nylon membranes, denatured and hybridized to a radiolabeled Telo2 probe as above. Radioactive signals were detected using a Typhoon FLA 9000 imager (GE Healthcare) and quantified using ImageJ software.

**Northern blotting**. Total RNA was isolated using the TRIzol reagent (Invitrogen) and treated three times with DNaseI (New England Biolabs). Fifteen micrograms of

RNA was separated on 1.2% agarose gels containing 0.7% formaldehyde. RNA was then transferred onto nylon membranes and hybridized to a radiolabeled Telo2 probe as above. Ethidium bromide (Sigma-Aldrich) stained tRNAs were used to control for loading. Radioactive signals were detected using a Typhoon FLA 9000 imager (GE Healthcare) and quantified using ImageJ software.

**Chromatin immunoprecipitation (ChIP)**. $10^6$ cells were harvested by scraping and resuspended in 1 ml of 1% formaldehyde at room temperature for 15 min. After quenching with 125 mM glycine, cells were washed three times in 1× PBS by centrifuging at $800 \times g$ for 5 min. Cell pellets were resuspended in 500 μl of lysis buffer (1% SDS, 50 mM Tris-HCl pH 8, 10 mM EDTA pH 8) supplemented with cOmplete Protease Inhibitor Cocktail (Roche) and sonicated twice using a Bioruptor apparatus (Diagenode) at 4 °C (settings: 30 s "ON" / 30 s "OFF"; power: "High"; time: 15 min). Cellular debris were pelleted by centrifugation at $1600 \times g$ at 4 °C for 10 min and 100 μl of supernatant were mixed with 1.1 ml of IP buffer (1% Triton X-100, 20 mM Tris-HCl pH 8, 2 mM EDTA pH 8, 150 mM NaCl). Diluted extracts were precleared by incubation with 50 μl of protein A/G-sepharose beads (GE Healthcare) blocked with sheared *E. coli* genomic DNA and BSA at 4 °C for 30 min on a rotating wheel, followed by centrifugation at $800 \times g$ at 4 °C for 5 min. Cleared extracts were incubated with 1 μg of anti-FANCM mouse monoclonal antibody (CE56.1,[72]) at 4 °C for 4 h on a rotating wheel. Immunocomplexes were isolated by incubation with blocked protein A/G beads at 4 °C overnight on a rotating wheel. Beads were washed four times with wash buffer 1 (0.1% SDS, 1% Triton X-100, 2 mM EDTA pH 8, 150 mM NaCl, 20 mM Tris-HCl pH 8) and once with wash buffer 2 (0.1% SDS, 1% Triton X-100, 2 mM EDTA pH 8, 500 mM NaCl, 20 mM Tris-HCl pH 8) by centrifuging at $800 \times g$ at 4 °C for 5 min. Beads were then incubated in 100 μl of elution buffer (1% Triton X-100, 20 mM Tris-HCl pH 8, 2 mM EDTA pH 8, 150 mM NaCl) containing 40 μg/ml RNaseA at 37 °C for 1 h, followed by incubation at 65 °C overnight to reverse crosslinks. DNA was purified using the Wizard SV Gel and PCR Clean-up kit (Promega), dot-blotted onto nylon membranes and hybridized overnight to a radiolabeled Telo2 probe as for TRF analysis. After signal detection membranes were stripped and re-hybridized overnight to radiolabeled Alu-repeat oligonucleotides as above. Radioactive signals were detected using a Typhoon FLA 9000 imager (GE Healthcare) and quantified using ImageJ software.

**DNA:RNA immunoprecipitation (DRIP)**. Cells were harvested by scraping and lysed in 1 ml of RA1 buffer (Macherey−Nagel) containing 1% v/v β-mercaptoethanol and 100 mM NaCl. Nucleic acids were extracted with phenol/chloroform/isoamyl alcohol (25:24:1 saturated with 10 mM Tris-Cl pH 7.0, 1 mM EDTA) and precipitated with isopropanol followed by centrifugation at $15,000 \times g$ at 4 °C for 10 min. Pellets were washed in 70% ethanol, resuspended in 200 μl Tris-EDTA, 100 mM NaCl and sonicated using a Bioruptor apparatus (Diagenode) at 4 °C (settings: 30 s "ON" / 30 s "OFF"; power: "High"; time: 5 min). Five micrograms of nucleic acids was incubated with 1 μg of S9.6 antibody (a kind gift from B. Luke, IMB, Mainz, Germany) in IP buffer (0.1% SDS, 1% Triton X-100, 10 mM HEPES pH 7.2, 0.1% sodium deoxycholate, 275 mM NaCl) at 4 °C for 5 h on a rotating wheel. For RNaseH control experiments, nucleic acids were incubated with 60 U of RNaseH in 1× RNaseH buffer or only with buffer at 37 °C for 3 h prior to incubation with the S9.6 antibody. Immunocomplexes were isolated by incubation with protein G Sepharose beads (GE Healthcare) blocked with sheared *E. coli* DNA and BSA. Beads were washed four times in IP buffer by centrifuging at $800 \times g$, and incubated in elution buffer (50 mM Tris-Cl pH 8, 10 mM EDTA, 0.5% SDS) containing 10 μg/ml proteinase K (Sigma-Aldrich) and 40 μg/ml RNase A at 50 °C for 30 min. Beads were centrifuged as above and supernatants recovered. Isopropanol-precipitated DNA was dot-blotted onto nylon membranes and hybridized to radiolabeled 5′-$(TTAGGG)_5$−3′ oligonucleotides as for TRF analysis. After signal detection membranes were stripped and re-hybridized to radiolabeled Alu-repeat oligonucleotides as for ChIP analysis. Radioactive signals were detected using a Typhoon FLA 9000 imager (GE Healthcare) and quantified using ImageJ software.

**In vitro R-loop resolution assays**. Flag-FANCM-8HIS:FAAP24 complex was purified using a baculovirus expression system in Sf9 cells. Cells were pelleted at $500 \times g$ and lysed on ice in 0.5 M NaCl, 0.02 M Triethanolamine pH 7.5, 1 mM DTT, 10% glycerol plus mammalian protease inhibitors (Sigma-Aldrich) and sonicated on ice $5 \times 10$ s bursts. Clarified lysates were incubated with equilibrated Flag M2 resin (Sigma-Aldrich) for 1 h and 4 °C. Flag resin was subjected to 5× batch washes and eluted with 100 μg/ml Flag peptide. Pooled FANCM-FAAP24 containing elutions were diluted to a final concentration of 100 mM NaCl, 20 mM TEA pH7.5, 10% glycerol, 1 mM DTT (Buffer B) and bound to 400 μl ssDNA affinity resin (Sigma-Aldrich). The resin was washed with 10 CV of buffer B. FANCM-FAAP24 complexes were eluted with buffer B containing 0.5 M NaCl. Two micrograms of pcDNA6-Telo or pcDNA6-TeloR plasmids containing a ~1 kb fragment of human telomeric repeats cloned downstream of a T7 promoter were in vitro transcribed using T7 polymerase (New England Biolabs) in presence of CTP, GTP, ATP (2.25 mM each) 825 nM [α-$^{32}$P]UTP (3000 Ci/mmol; Perkin Elmer). Reactions were stopped by heating to 65 °C for 20 min followed by RNase A (EpiCentre) treatment in 330 mM NaCl. R-loop-containing plasmids were

purified by two phenol:chloroform extractions. Unincorporated nucleotides were removed by passing nucleic acids twice through S-400 columns (GE Healthcare). R-loop unwinding reactions (10 μl final volume) contained 1 nM R-loop plasmids, 1 mM ATP, 2.5 nM FANCM-FAAP24 in R-loop buffer (6.6 mM Tris pH 7.5, 3% glycerol, 0.1 mM EDTA, 1 mM DTT, 0.5 mM $MgCl_2$). Reactions were performed at 37 °C for 10 min and then stopped by adding 2 μl of stop buffer (10 mg/ml proteinase K (New England Biolabs), 1% SDS) and incubating at 37 °C for 15 min. Samples were run on 0.8% agarose TAE gels in TAE buffer (40 mM Tris, 20 mM acetic acid, 1 mM EDTA) at 100 V for 60–90 min, followed by gel drying and autoradiography.

**Statistical analysis**. For direct comparison of two groups, we employed a paired two-tailed Student's *t* test using Microsoft Excel or a nonparametric two-tailed Mann−Whitney *U* test using GraphPad Prism. For comparison of two or more factors for each group and their interaction, we used a two-way analysis of variance (ANOVA) followed by Tukey's HSD for the pairwise comparisons. The analysis was carried out using the aov and TukeyHSD functions of R version 3.3.2. The significance levels are from the Tukey's HSD adjusted $P$ values. $P$ values are indicated as: *$P < 0.05$, **$P < 0.005$, ***$P < 0.001$, ****$P < 0.0001$.

**Reporting summary**. Further information on research design is available in the Nature Research Reporting Summary linked to this article.

## Data availability
The data that support the findings of this study are available from the corresponding author upon reasonable request.

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

## Acknowledgements

We thank M. Lopes, B. Fuchs, A. Londoño-Vallejo, O. Shakhova, K. Collins, M. Carmo-Fonseca, T. de Lange, and B. Luke for reagents; Filipe Tavares-Cadete for help with statistical analysis; Patricia Abreu, José Nobre and Patrick Stalder for help with preliminary experiments; the Bioimaging and Flow Cytometry facilities of iMM and the Scientific Center for Optical and Electron Microscopy of ETHZ for services. Research in the Azzalin laboratory was supported by the Swiss National Science Foundation (31003A_160338), the European Molecular Biology Organization (IG3576) and the Fundação para a Ciência e a Tecnologia (IF/01269/2015; PTDC/MED-ONC/28282/2017; PTDC/BIA-MOL/29352/2017). R.P. was supported by a Swiss National Science Foundation Doc.Mobility fellowship

(P1EZP3-168771). Research in the Deans laboratory was supported by the Cancer Council of Victoria, Australian National Health and Medical Research Council (APP1139099), Buxton trust and the Victorian Government's OIS Program. A.J.D is a Victorian Cancer Agency fellow. Publication costs were supported by UID/BIM/50005/2019, project funded by the Fundação para a Ciência e a Tecnologia/Ministério da Ciência, Tecnologia e Ensino Superior (MCTES) through Fundos do Orçamento de Estado.

## Author contributions

R.P. and C.M.A. conceived the original project. B.S., R.P., R.A. and C.M.A designed experiments. B.S., R.P., R.A., A.M.F., H.W. and C.M.A. performed the experiments and data analysis. Y.W.L. performed the DRIP experiments. C.H. and A.J.D. designed and performed the in vitro R-loop resolution assays. A.J.D. generated the anti-FANCM monoclonal antibody. C.M.A wrote the manuscript with inputs from A.J.D., B.S., R.P. and R.A.

## Additional information

**Competing interests:** The authors declare no competing interests.

