## [Peer Review File · Nature Communications]

Reviewers' comments:

Reviewer #1 (Remarks to the Author):

Pentz et al evaluate the function of Fanconi Anemia M (FANCM) protein at telomeres in cells that activate the ALT mechanism of telomere elongation. To do so, the authors conducted an analysis of phenotypic markers of ALT activity – particularly C circles- partially single stranded C-rich extra-chromosomal telomeric DNA species found almost exclusively in ALT cancer cells. They present pretty extensive data showing that FANCM depletion leads to increased production of C-circles, increased APBs and disturbing the balance of anti and pro- recombinogenic stimuli that underpin the ALT mechanism. They show that this is dependent on the BLM helicase implicating FANCM as suppressing BLM activity at ALT telomeres. Furthermore, an accumulation of telomeric R-loops is also detected that appears to correlate with the accumulation of C-circles. These form RNA:DNA hybrids that exacerbate the instability of ALT cells – driving them to catastrophic outcomes.

The paper presents lots of data – some of which needs to be tidied up. The paper is lacking on a clear link between the evident death/arrest of ALT cells and the absence of telomere shortening and mechanistic insights as to how FANCM suppresses CC formation. Strengthening the evidence for a direct link with telomeres would be positive, since most of the effects occur in ALT cells in which the cells accumulate in G2. So this may be the reason for cell death and not precisely due to telomere dysfunction, as is implied. Below are a series of experiments that could be done to improve the paper and provide some necessary novelty to the paper.

Major issues.

- It would strengthen the paper if a rescue test experiment should be conducted and CCs, APBs etc could be tested and also to what activity of FANCM is required to suppress CC/BLM toxicity.
- Show FANCM localization by IF with antibody or tagged proteins. Mechanism of recruitment should be evaluated/addressed since it evidently is not via BLM. This could be linked to the above experiments.
- G2 arrest experiments should be re-evaluated (see below).

Other issues to be addressed.

- Page 4. Not sure there can be progressive telomere loss without telomere shortening. Suggest amend statement as it could be misleading.
- Figure 1. No validation of TERT/hTR expression is shown
- Figure 2C. ChIP section can be interpreted to imply there is more FANCM at Alu repeats arguing against ALT specificity. If I have misunderstood the figure, I think it would help to present in an alternative manner.
- Figure 3.
 - o Use of TRF2 as a surrogate of telomeres in G2 is maybe incorrect - based on evidence that TRF2 is displaced from telomeres in G2 (Verdun & Karlseder, Hayashi et al, Mol Cell, Cell, NSMB). Suggest FISH throughout this section.
 - o Cells do not look like G2 cells. Usually MIDAS is assessed in prometaphase cells (Hickson). Especially if the cells were synchronized and then released. After 2.5hrs they should be in mitosis – this does not seem to be the case.
 - o Synchronization with CDKi does not seem to have worked very well – even in the control cells. Greenberg/Hickson show a minimum 70-90% synchronization in U2OS cells. Release seems to have little effect? If anything there are more cells in S-phase.

- o For studies of nascent G2-mitotic DNA synthesis, which is inferred in the text – EdU or BrdU should be used in combination with TTAGGG FISH.
- No TFEs are detected, per the – but increased RAD51 is observed in Figure 4. Have TSCes been examined.
- Figure 5. The authors focus on CCs. These are one species of
- Figure S1. Label the sections please.
- Figure S2. Data on clustering seems at odds with the statement that FANCM siRNA leads to greatly enhanced clustering. Yes, I see the statistics but the difference is really subtle as presented in S2D. Suggest alternative presentation or perhaps testing effect in TRF1-FokI system (Greenberg Lab) to be more convincing.
- Figure S2C. Show nuclear/telomeric mask or DAPI for clarity.

Reviewer #2 (Remarks to the Author):

In this manuscript the authors investigate the role of FANCM in the Alternative Lengthening of Telomeres pathway. Here, the authors demonstrate that FANCM depletion leads to a significant increase in replication stress at ALT telomeres, which results in the accumulation of DNA damage and ultimately, G2/M cell cycle arrest. To further define FANCM function at ALT telomeres, the authors investigated how replication stress was induced following FANCM depletion. Supporting previous in vitro work in the field, the authors demonstrate the FANCM is able to resolve R-loop structures. In this manuscript, the authors show that R-loops accumulate at telomeres when FANCM is depleted by using DNA-RNA immunoprecipitation, pointing to FANCM as an R-loop resolving enzyme at ALT telomeres. The authors conclude that FANCM depletion leads to persistent TERRA R-loops, and replication stress, which culminate in the exacerbation of ALT phenotypes. Finally, the authors demonstrate that depletion of the BLM helicase alleviates the replication stress induced by FANCM loss suggesting that unchecked BLM is toxic in ALT cells. Overall, this manuscript provides compelling evidence for an important function of FANCM in mitigating replication stress at ALT telomeres, and contributes novel mechanistic information on how FANCM accomplishes this through resolving R-loops. However, there are both major and minor concerns to be addressed.

Major concerns

- In the title of the manuscript the authors state ‘FANCM assures regulated ALT activity by restricting BLM- and R-loop-induced telomeric replication stress’ in addition in the abstract the authors write ‘FANCM allows controlled ALT activity and ALT cell proliferation by limiting the toxicity of BLM and telomeric R-loops’. These statements imply that BLM itself is inducing replication stress. This conclusion appears to be drawn from the co-depletion experiments in Figure S4 and Figure 6. To further support such an assertive conclusion the authors should use shRNA or CRISPR to co-deplete FANCM and BLM and culture these cells long term to demonstrate that the cells are in fact viable long term. Would these data also argue that BLM depletion would rescue FANCM deficient cells exposed to exogenous replication stress as well? The long term viability assays, would further support the siRNA experiments and bolster the main conclusion of the paper.
- Likewise, The conclusion that FANCM ‘alone is essential for ALT cell viability’ (bottom of page 10) is overstated simply based on the data presented. All of the assays are proliferative assays and do not measure viability itself. Moreover, throughout the rest of the paper the authors make the argument that FANCM depletion leads to a G2/M arrest. Given that ALT telomeres experience elevated levels of replication stress at the telomeres is it possible that FANCM depletion would lead to G2/M arrest in any cell experiencing chronic replication stress and not just ALT cells. This is one of the defining features of FA, they accumulate in G2/M in the presence of replication stress induced by ICLs. The ICLs promote replication stress and lead to checkpoint response at the G2/M boundary. If you were to use a cell line that is known to experience elevated levels of replication stress would loss of FANCM induce G2/M arrest here too (i.e. T98G)? Longterm what happens to

the cells? The authors show the colony formation assays, but clearly there are still cells growing, do they eventually die?

- In Figure 2C the authors state that “FANCM-mediated suppression of telomere instability is likely to be direct, because the protein was found associated with telomeric DNA in chromatin immunoprecipitation (ChIP) experiments using anti-FANCM antibodies (Fig. 2C and D)”. The argument is that FANCM is doing something distinct at ALT telomeres to mitigate replication stress and that when this function is defective this leads to genome instability and G2/M arrest. However, the data also show that FANCM binds to Alu repeats, and in fact seems to bind Alu repeats better than telomeres? How does FANCM binding to telomeric repeats compare to a known telomere binding protein, such as TRF1 or TRF2? If the argument is that FANCM binds the ALT telomeres more efficiently than non-ALT telomeres, the authors should also include ChIP with the H-ST cell line? That being said, is the result that FANCM binds Alu repeats more efficiently than telomeres accurate? What does this say about potential function of FANCM in the maintenance of Alu repeats? Do they undergo replication stress that is also mitigated by FANCM? This needs to be addressed.
- The changes in telomeric foci size are impressive and while perhaps not in every cell they are clearly prevalent. However, the statistical quantification of the data as presented in Figure S2E, trivializes the result given that the standard deviations are, in several cases, equal to or larger than the mean itself. Would it be useful to instead, or in addition, quantify the percentage of cells with the larger foci? I.e. % of cells with foci size greater than 70 pixels for example and include stats there? Perhaps even add that to Figure 4? It just seems like the authors are underselling this piece.
- Figure 3 and 4 seem to be making a similar conclusion, that FANCM depletion leads to an increase in APB and EdU incorporation at ALT telomeres. This is not simply a product of G2/M arrest in the FANCM depleted cells as control cells arrested in G2/M with RO-3306 do not induce the same phenotypes. These two figures should be combined and condensed.
- The telomere foci following FANCM depletion are significantly larger when measured by telomere FISH as compared to TRF2 immunofluorescence, is that true or potentially just the images selected?
- In Figure 5, the authors detect an increase in C-rich single stranded DNA after FANCM depletion. The results presented in 5B are quite nice, but 5A appears to demonstrate any phenotype with VA13. Perhaps VA13 just has more robust phenotypes in the absence of FANCM, which is fine, the authors should just include VA13 in the data presented in 5B for consistency.
- In Figure 7 it appears that FANCM unwind C-rich R-loops better than G-rich? Does this mean that FANCM can resolve TERRA R-loops, but perhaps not as efficiently as another factor?
- If FANCM depleted cells induce replication stress at telomeric DNA, isn't it possible that the S9.6 antibody is also detecting okazaki fragments present at stalled replication forks? So in addition to R-loops you are also detecting Okazaki fragments? Perhaps treat your DRIP experiments with RNASEH2 in addition to RNASEH1? This would further support the model that you have increased replication stress in which some is due to R-loops at telomeric DNA, but some is also due to other structures. How do levels of TERRA in U2OS compare to other ALT lines, is the FANCM phenotype blunted in cells that potentially have lower TERRA expression?
- While the authors have done their diligence to include statistical analysis throughout the paper, the tests do not always seem to be appropriate for the experiment being analyzed and as a result some of the significance may be overestimated? The authors should consult a biostatistician to support their analysis.

Minor revisions

- The figures contain a multitude of abbreviations, and while they are all defined in the figure legends, it makes the figures much harder to interpret and understand.
- Figure S1 needs panel identifiers (A, B, C etc.) in order to directly reference the relevant cell cycle profile for experiments in the text.
- In Figure 3 the authors state that FANCM depletion increases telomeric DNA synthesis outside of S-Phase, however the quantification of relative cell count for each phase of the cell cycle in Figure

3C demonstrates that there are still approximately 40-60% of cells in G1/S. How are the S-phase cells excluded from the analysis in Figure 3A-B? The cells with greater than 5-foci, but less than X foci? 10? 20? Just a little clarity on the analysis would be helpful here.

- The quantification of ECTR by FISH in Figure 5 should be better detailed. The authors should describe in words what was considered a positive ECTR event?
- The text incorrectly references Figure 5A as Figure 5D (sentence beginning "When hybridization was performed under native...")
- In Figure 5E, it is unclear whether the U2OS and VA13 data is not significant, or whether statistics were not performed on these data.
- Figure 7A lacks quantification.
- There are not representative image for all of the conditions quantified in Figure 8B

Reviewer #3 (Remarks to the Author):

In their manuscript entitled "FANCM assures regulated ALT activity by restricting BLM- and R-loop-induced telomeric replication stress", Penz et al have demonstrated the essential function of FANCM, specifically at telomeres that are being maintained through the ALT mechanism. Indeed they have convincingly shown that many ALT features are strongly exaggerated with FANCM loss, including c-circle accumulation, telomere replication stress and APB formation. They suggest that although replication stress is required, to an extent, for the BIR mediated repair of ALT telomeres, it is kept in check by FANCM to ensure continued viability. The authors demonstrate that the source of the replication stress is due, in part, to the accumulation of TERRA R-loops at telomeres. Accordingly, the expression of RNase H alleviates many of the siFANCM telomeric phenotypes. It is suggested, that particularly in the case of ALT cancers, FANCM may be an effective target for cancer therapy as it seems to specifically target ALT cells and not telomerase positive cells. In conclusion, the manuscript is well written and the data depicted are solid. It is an important study that has important implications. I support its publication but do have a few points that I would like to see addressed.

Although the manuscript strongly suggests that the loss of viability in ALT cells is due to the mode of telomere maintenance (for good reason) it could potentially be independent of telomeres. I feel that one important control would be to test if the loss of ATRX (mutated in 95% of ALT cell lines) is synthetic lethal with loss of FANCM in telomerase positive cells.

I was surprised to see that the expression of "super telomerase" did not rescue the lethality of U2OS cells, which would suggest that the ALT telomerase maintenance mode might not be responsible for this genetic interaction. Can the authors please comment (or demonstrate) what actually happens when telomerase is overexpressed in ALT cells on the level of c-circles, APBs and TERRA levels/R-loops. Doesn't the "telomerase mode" of elongation take over, as is the case in the budding yeast system, and how can we be sure that telomerase is actually acting on telomeres in the U2OS cells.

It is suggested that FANCM inhibition may be beneficial for ALT related cancer therapy based on the fact that the inhibition of FANCM was not toxic in telomerase positive cells. However, since 80% of the cells in humans are telomerase negative, what are the effects in telomerase negative cells? Or in cells where telomerase has been pharmacologically or genetically inhibited?

Minor comments:

In general, in a many of the figures it would help if the ALT cell lines and Telomerase (+) cell lines were indicated directly on the figure in some manner, maybe it's not required in all figures but at least in figure 1. Moreover, in the discussion it would be useful to refer to the relevant figures.

p.9: 'Figure 5D, upper panel'. There is no upper panel for 5D. It should probably say '5A'

'These results establish that FANCM and BLM depletion are not synthetically lethal in ALT cells, and rather reveal a synthetic viable interaction'. I think this should be toned down accordingly as I feel that synthetic lethals are difficult to conclude with incomplete knock down experiments using siRNA.

p.12: The FISH applied in 7D could be more clearly explained. Maybe a cartoon would help to demonstrate why more C-strand is accessible to the probe following RNase H treatment.

p.13 top: 'we propose that FANCM directly resolves telR-loops on telomeric chromatin'. This does not explain why single stranded C-rich DNA accumulates in siFa cells in the absence of RNaseH (Figure 7D and E). Could this be due to replication gaps?

In the discussion it could be mentioned that the Aguilera lab has recently demonstrated that Mph1 is recruited to short telomeres in an RNase H sensitive manner (Lafuente-Barquero et al, 2017).

Point-by-point response to the Reviewers' comments

Please note that Reviewers' comments are in blue italics and our answers have been numbered to facilitate cross-referencing.

Reviewer #1 (Remarks to the Author):

Pentz et al evaluate the function of Fanconi Anemia M (FANCM) protein at telomeres in cells that activate the ALT mechanism of telomere elongation. To do so, the authors conducted an analysis of phenotypic markers of ALT activity – particularly C circles- partially single stranded C-rich extra-chromosomal telomeric DNA species found almost exclusively in ALT cancer cells. They present pretty extensive data showing that FANCM depletion leads to increased production of C-circles, increased APBs and disturbing the balance of anti and pro-recombinogenic stimuli that underpin the ALT mechanism. They show that this is dependent on the BLM helicase implicating FANCM as suppressing BLM activity at ALT telomeres. Furthermore, an accumulation of telomeric R-loops is also detected that appears to correlate with the accumulation of C-circles. These form RNA:DNA hybrids that exacerbate the instability of ALT cells – driving them to catastrophic outcomes.

The paper presents lots of data – some of which needs to be tidied up. The paper is lacking on a clear link between the evident death/arrest of ALT cells and the absence of telomere shortening and mechanistic insights as to how FANCM suppresses CC formation. Strengthening the evidence for a direct link with telomeres would be positive, since most of the effects occur in ALT cells in which the cells accumulate in G2. So this may be the reason for cell death and not precisely due to telomere dysfunction, as is implied. Below are a series of experiments that could be done to improve the paper and provide some necessary novelty to the paper.

Answer #1

We thank this Reviewer for appreciating our work and the amount of data that were presented. We agree that the first version of our manuscript did not address clearly the connection between telomere instability and cell cycle proliferation defects in FANCM-depleted cells. While it is true that in FANCM-depleted ALT cells telomeres do not appear to shorten (which is possibly due to the fact that cells are harvested only 2 days after transfection), they show signs of profound instability, including strong accumulation of pS33 and 53BP1. Therefore, although shortening is not detected (yet), the DNA damage signal emanating from replicatively stressed telomeres appears to be strong enough to impair proper proliferation and trigger cells death, as it is now shown in Fig. 1F and Fig. S2A. Additionally, we have now depleted FANCM in U2OS cells over-expressing the shelterin factor TRF1, a condition that we previously reported to halve the incidence of fragile telomeres, possibly because TRF1 is limiting in ALT cells¹. FANCM depletion in TRF1 over-expressing cells still led to G2/M accumulation and RPA phosphorylation at serine 33, yet less severely than in control cells. We conclude that telomeric replication stress contributes to the sensitivity of ALT cells to FANCM depletion. This is shown in Fig. 1 G and H and discussed in the text.

Major issues.

- *It would strengthen the paper if a rescue test experiment should be conducted and CCs, APBs etc could be tested and also to what activity of FANCM is required to suppress CC/BLM toxicity.*

Answer #2

We have now performed complementation experiments in U2OS cells infected with retroviruses ectopically expressing an siRNA-resistant, V5 epitope-tagged FANCM variant (V5-FANCM WT) or an ATPase/translocase dead counterpart unable to resolve R-loops (V5-FANCM K117R²). Confirming the specificity of our siRNAs, V5-FANCM WT expression largely averted G2/M arrest and accumulation of ps33 TIFs and APBs in FANCM-depleted cells (Fig. 8B and C; Fig. S1D; Fig. S7A). On the contrary, V5-FANCM K117R failed to rescue the same defects, which remained at levels comparable to the ones seen in FANCM-depleted control cells (Fig. 8B and C; Fig. S1D; Fig. S7A). We therefore show that the ATPase/translocase activity of FANCM is essential to maintain regulated ALT activity and cell proliferation. This further strengthens our conclusion that deregulated telR-loops trigger replication stress in ALT cells depleted of FANCM. Moreover, these data are consistent with the ones described in the accompanying manuscript 'The FANCM-BLM-TOP3A-RMI complex suppresses alternative lengthening of telomeres (ALT)' (Lu et al., Pickett laboratory). The authors found that contrarily to FANCM WT, FANCM K117R did not suppress ALT features when over-expressed. They also characterized a large number of other mutants and found that not only FANCM translocase activity but also FANCM interaction with the BLM-TOP3A-RMI complex is crucial to regulate ALT. We believe that our data and the ones from the Pickett laboratory complement and strengthen each other very well. After consulting with the editor, we decided not to perform complementation experiments with additional mutants in order to avoid too extensive redundancy between the two manuscripts.

- *Mechanism of recruitment should be evaluated/addressed since it evidently is not via BLM. This could be linked to the above experiments.*

Answer #3

How FANCM is recruited to telomeres in ALT cells is indeed a very interesting question. Contrarily to what was previously reported³, we have not been able to detect accumulation of FANCM at telomeres in ALT cells by indirect immunofluorescence (IF), using 6 different antibodies. Rather, we have always detected a diffuse, pan nuclear staining, which was increased upon cell treatment with hydroxyurea (HU) or aphidicolin (Fig. R1). Similarly, staining with anti-V5 antibodies of U2OS cells expressing ectopic FANCM failed to reveal a punctate, telomeric staining (not shown). This data are consistent with the fact that endogenous FANCM is found exclusively in chromatin fractions from different cell types⁴. A straightforward way to analyze FANCM recruitment specifically to telomeres is not available, and would require laborious ChIP experiments as the ones we presented in Fig. 2C and D. We feel that these experiments go beyond the scope of the current manuscript, which already presents a large amount of data. Nevertheless, as pointed out by Reviewer #3, it has been recently demonstrated that Mph1, the *Saccharomyces cerevisiae* ortholog of human FANCM, localizes to short, dysfunctional telomeres in an R-loop-dependent manner, as its association with short

telomeres was impaired when RNaseH1 was over-expressed⁵. It is therefore likely that telR-loops, which are abundant at ALT telomeres, contribute to FANCM recruitment and/or stabilization. We now mention this in the Discussion.

Figure R1. Examples of anti-FANCM and anti-TRF2 IF experiments performed in U2OS and HeLa cells treated with 0.2 mM hydroxyurea (HU) for or 0.2 μ M aphidicolin (APH) for 18 hours or left untreated. A rabbit polyclonal anti-FANCM (Bethyl Laboratories, A302-637A; in red) and a mouse monoclonal anti-TRF2 (Millipore, 05-521; in green) were used. In the merge panel, DAPI stained DNA is shown in blue. Cells were permeabilized before fixation in order to eliminate soluble proteins not bound to chromatin. Note the diffuse pan-nuclear staining for FANCM and the increase in signal intensity observed in some cells upon drug treatment.

- *G2 arrest experiments should be re-evaluated (see below).*

Other issues to be addressed.

- *Page 4. Not sure there can be progressive telomere loss without telomere shortening. Suggest amend statement as it could be misleading.*

Answer #4

We have amended the sentence and we now say 'RNaseH1 over-expression causes progressive TFE accumulation'.

- *Figure 1. No validation of TERT/hTR expression is shown.*

Answer #5

We have performed quantitative RT-PCR experiments and we now show that both hTERT and hTR RNAs are heavily over-expressed in HeLa and U2OS supertelomerase cells. We also show

that, as previously reported^{6,7}, HeLa supertelomerase cells have considerably elongated telomeres and that supertelomerase U2OS cells carry less telomere free ends, although telomere fragility is not affected. Data are shown in Fig. S3 and discussed in the text.

• *Figure 2C. ChIP section can be interpreted to imply there us more FANCM at Alu repeats arguing against ALT specificity. If I have misunderstood the figure, I think it would help to present in an alternative manner.*

Answer #6

We do not intend to imply that FANCM binds only to telomeres in ALT cells. As mentioned above, FANCM is found in chromatin fractions of several cell types⁴, and our IF experiments show a diffuse nuclear staining in U2OS cells (Fig. R1). We therefore believe that FANCM associates with chromatin at large, and the detected binding to Alu repeat DNA supports this assumption. Structures such as R-loops or D-loops might further recruit, stabilize or activate FANCM at specific chromatin loci, including telomeres. We have explained this in the Results and Discussion sections.

• *Figure 3.*

o Use of TRF2 as a surrogate of telomeres in G2 is maybe incorrect - based on evidence that TRF2 is displaced from telomeres in G2 (Verdun & Karlseder, Hayashi et al, Mol Cell, Cell, NSMB). Suggest FISH throughout this section.

Answer #7

We have quantified the number of TRF2 foci and their area in FANCM-depleted U2OS cells and compared them with the ones of foci detected by telomeric DNA FISH. As shown in Fig. R2, it is true that the overall area of TRF2 foci is smaller than the one of telomeric DNA foci, and the increase observed upon FANCM depletion is less important when anti-TRF2 IF is used. This might indeed be due to decrease binding of TRF2 to telR-loop-containing telomeres, or both. Nevertheless, the numbers of TRF2 and telomeric foci are very similar as are the changes observed upon FANCM depletion. We conclude that TRF2 remains a good surrogate for telomere foci even in FANCM-depleted cells, as long as only their number is considered. We have anyway repeated the EdU incorporation experiments and used telomeric FISH instead of anti-TRF2 IF (Fig. 4; see also Answer #8 below). Additionally, we have performed anti-POLD3 IF experiments and we now show that POLD3 accumulates at telomeres in FANCM-depleted U2OS cells (Fig. 4), further corroborating the idea that FANCM suppresses BIR-mediated telomere synthesis outside of S-phase. Please note that we could not perform IF combined with DNA FISH due to loss of POLD3 signal when a denaturation step was included in the procedure. We therefore resorted to double IF for POLD3 and the TRF2-associated telomeric marker RAP1.

Figure R2. Quantifications of numbers and areas of foci detected using telomeric DNA FISH and anti-TRF2 IF in U2OS cells transfected with the indicated siRNAs and harvested 48 hours after transfection. Each dot represents an individual nucleus. A total of at least 300 nuclei from three independent experiments were analyzed for each sample. Bars and error bars are means and SDs. Note that both protocols detect similar numbers of foci, while areas appear to be larger in FISH samples than in IF samples.

o Cells do not look like G2 cells. Usually MIDAS is assessed in prometaphase cells (Hickson). Especially if the cells were synchronized and then released. After 2.5hrs they should be in mitosis – this does not seem to be the case.

o Synchronization with CDKi does not seem to have worked very well – even in the control cells. Greenberg/Hickson show a minimum 70-90% synchronization in U2OS cells. Release seems to have little effect? If anything there are more cells in S-phase.

o For studies of nascent G2-mitotic DNA synthesis, which is inferred in the text – EdU or BrdU should be used in combination with TTAGGG FISH.

Answer #8

In our hands RO-3306 treatment of U2OS cells leads to ~ 65 to 79 % of cells arrested in G2/M phase. Fig. R3 shows results from some of the different experiments utilized for this study. Importantly, the fraction of G2/M cells obtained by blocking siCt-transfected cells is never below the ones in cells depleted for FANCM (Fig. R3). We now show more representative PI/FACS quantifications in Fig. 3B. Additionally, as shown in the previous version of this manuscript, RO-3306-treated, FANCM-depleted cells do not progress out of G2/M upon drug wash-off. To avoid any confounding aspects associated to uneven block/release, we have now repeated all experiments to detect nascent DNA synthesis using siCt cells either untreated or blocked in G2/M using RO-3306, and FANCM-depleted cells. Moreover, we have used telomeric DNA FISH (instead of anti-TRF2 IF) in combination with EdU detection, and we have not scored cells with pan nuclear EdU staining or more than 25 EdU foci, to exclude S-phase cells from the analysis. These experiments, presented in Fig. 4A and B, confirm that FANCM depletion suppresses telomeric DNA synthesis outside of S phase.

Figure R3. Examples of PI/FACS profiles of U2OS cells transfected with the indicated siRNAs and harvested 48 hours after transfection. For G2/M Block, siCt-transfected cells were treated with 10 μ M RO-3306 for 18 (first 4 samples from the left) or 24 (last two samples) hours. The fraction (%) of cells in G2/M are indicated in red for each sample. Note that the fraction of G2/M cells in RO-3306-arrested cells is never below the ones in cells depleted for FANCM.

- *No TFEs are detected, per the – but increased RAD51 is observed in Figure 4. Have TSCes been examined.*

Answer #9

We have tried several times to preform CO-FISH experiments in ALT cells depleted for FANCM, yet unsuccessfully, as we have never been able to efficiently detect telomeric hybridization signals after UV/ExoIII treatment. We do not know why this happens but it might have to do with inefficient incorporation of BrdU/BrdC or the presence of nicked telomeric DNA in FANCM-depleted cells. This has impeded the analysis of TSCes, although we recognize that it would be an interesting point to follow up. We now mention in the Discussion that the detected accumulation of RAD51 at telomeres in FANCM-depleted cells suggests that ‘RAD51 could partly mediate the telomere clustering observed in FANCM-depleted cells or other events which we did not explore, such as for example sister telomere exchanges’.

- *Figure 5. The authors focus on CCs. These are one species of*

Answer #10

Unfortunately the sentence seems to be incomplete; nevertheless we believe that the Reviewer wanted to mention that CCs are not the only species of ECTRs detected in ALT cells. In the Discussion section we have now clarified that ‘FANCM suppresses ALT-associated features, including... production of ECTRs which include C-circles and possibly other forms’.

- *Figure S1. Label the sections please.*

Answer #11

The figure has been labeled and appropriately referenced throughout the text.

- *Figure S2. Data on clustering seems at odds with the statement that FANCM siRNA leads to greatly enhanced clustering. Yes, I see the statistics but the difference is really subtle as presented in S2D. Suggest alternative presentation or perhaps testing effect in TRF1-FokI system (Greenberg Lab) to be more convincing.*

Answer #12

Following the suggestion of this Reviewer and of Reviewer #2, we have now added quantifications of cells with at least 5 large foci with an area of 60 pixels or larger. Approximately 60% of FANCM-depleted cells were scored as positive, versus approximately 10% and 15% of untreated or RO-3306-treated siCt-transfected cells, respectively. Those data are presented in Fig. 3F and described in the text. Please note that the original Fig. S2 is now main Fig. 3.

- *Figure S2C. Show nuclear/telomeric mask or DAPI for clarity.*

Answer #13

We have added white dotted lines to outline nuclei. Please note that the original Fig. S2 is now main Fig. 3.

Reviewer #2 (Remarks to the Author):

In this manuscript the authors investigate the role of FANCM in the Alternative Lengthening of Telomeres pathway. Here, the authors demonstrate that FANCM depletion leads to a significant increase in replication stress at ALT telomeres, which results in the accumulation of DNA damage and ultimately, G2/M cell cycle arrest. To further define FANCM function at ALT telomeres, the authors investigated how replication stress was induced following FANCM depletion. Supporting previous in vitro work in the field, the authors demonstrate the FANCM is able to resolve R-loop structures. In this manuscript, the authors show that R-loops accumulate at telomeres when FANCM is depleted by using DNA-RNA immunoprecipitation, pointing to FANCM as an R-loop resolving enzyme at ALT telomeres. The authors conclude that FANCM depletion leads to persistent TERRA R-loops, and replication stress, which culminate in the exacerbation of ALT phenotypes. Finally, the authors demonstrate that depletion of the BLM helicase alleviates the replication stress induced by FANCM loss suggesting that unchecked BLM is toxic in ALT cells. Overall, this manuscript provides compelling evidence for an important function of FANCM in mitigating replication stress at ALT telomeres, and contributes novel mechanistic information on how FANCM accomplishes this through resolving R-loops. However, there are both major and minor concerns to be addressed.

Answer #1

We thank the Reviewer for recognizing the importance of our work as it helps better understand not only how FANCM works mechanistically, but also how ALT cancer cells proliferate. We fully agree with the Reviewer's statement that 'unchecked BLM' is toxic in ALT cells', and we have taken this mention on board throughout the entire text in the revised manuscript. Indeed, we do not want to necessarily imply that BLM *per se* is toxic, but rather that it becomes toxic when is not properly regulated by FANCM (see also below). This is also mentioned in the new title of our manuscript: 'FANCM assures regulated ALT activity by restricting telomeric replication stress induced by deregulated BLM and R-loops'.

Major concerns

• *In the title of the manuscript the authors state 'FANCM assures regulated ALT activity by restricting BLM- and R-loop-induced telomeric replication stress' in addition in the abstract the authors write 'FANCM allows controlled ALT activity and ALT cell proliferation by limiting the toxicity of BLM and telomeric R-loops'. These statements imply that BLM itself is inducing replication stress. This conclusion appears to be drawn from the co-depletion experiments in Figure S4 and Figure 6. To further support such an assertive conclusion the authors should use shRNA or CRISPR to co-deplete FANCM and BLM and culture these cells long term to demonstrate that the cells are in fact viable long term. Would these data also argue that BLM depletion would rescue FANCM deficient cells exposed to exogenous replication stress as well? The long term viability assays, would further support the siRNA experiments and bolster the main conclusion of the paper.*

Answer #2

As mentioned above, we have now toned down the statement that BLM is toxic; we now state that BLM becomes toxic in ALT cells when deregulated in absence of FANCM. We believe that this statement is correct as we demonstrate that depleting BLM alleviates the molecular and cellular defects promoted by FANCM depletion. Moreover, this statement implies that regulated BLM assures proper ALT activity, which is consistent with previous work from the Pickett laboratory showing that over-expression of BLM increases ALT features including APBs and C-circles, while BLM depletion decreases ALT⁸.

As for the long term culturing experiments, we have now performed FANCM depletion using lentivirus-mediated stable expression of shRNAs. As shown in Fig. R4, shRNA-mediated depletion was not as efficient as with siRNAs (compare depletion in U2OS in Fig. R4A with the ones showed in the manuscript). Similarly, FANCM depletion-associated defects (pS33 and G2/M cell accumulation) were milder in shRNA-infected U2OS cells than in siRNA-transfected ones (compare pS33 western blot in Fig. R4A and propidium iodide (PI)/FACS profiles in Fig. R4B with the ones showed in the manuscript). This is not surprising, as in our hands lentiviral shRNAs often deplete less efficiently than siRNAs. Additionally, counter-selection against too strong FANCM depletion might occur in U2OS cells stably infected with shRNAs. For these reasons, we were unable to perform the suggested long-term experiment. However, we have performed growth curve experiments over a time period of 10 days and by re-transfecting cells with siRNAs every three days. We now clearly show that U2OS cells (but not HeLa cells) depleted for FANCM are quickly eliminated from the population (Fig. 1F). This elimination is likely due to cell death because FANCM depleted U2OS cells (but not HeLa cells) start to become permeable to PI already at 3 days after siRNA transfection (Fig. S2A). As for BLM depletion, this also led to an impaired cell growth in U2OS cells (Fig. 6D), yet in absence of PI permeabilization (Fig. S2A), suggesting lower proliferation rates rather than cell death. Importantly, BLM depletion partly averted proliferation defects (Fig. 6D) and death (Fig. S2A) in U2OS cells depleted for FANCM. These new data, which are fully consistent with our colony forming assays, further indicate that the proliferation defects associated to FANCM depletion depends on the presence of BLM.

Figure R4. Lentiviral shRNA-mediated depletion of FANCM in the indicated cell lines. MRC5 and HLF are primary human lung fibroblasts. Cells were harvested 8 days after lentiviral infections. **(A)** Western blot analysis of FANCM and pS33. Lamin B1 (LMB1) and Golgin serve as loading controls. **(B)** Examples of PI/FACS profiles. Note that FANCM depletion did not lead to pS33 and G2/M cell accumulation in HeLa cells nor in primary fibroblasts

• Likewise, The conclusion that FANCM ‘alone is essential for ALT cell viability’ (bottom of page 10) is overstated simply based on the data presented. All of the assays are proliferative assays and do not measure viability itself. Moreover, throughout the rest of the paper the authors make the argument that FANCM depletion leads to a G2/M arrest. Given that ALT telomeres experience elevated levels of replication stress at the telomeres is it possible that FANCM depletion would lead to G2/M arrest in any cell experiencing chronic replication stress and not just ALT cells. This is one of the defining features of FA, they accumulate in G2/M in the presence of replication stress induced by ICLs. The ICLs promote replication stress and lead to checkpoint response at the G2/M boundary. If you were to use a cell line that in addition, we have is known to experience elevated levels of replication stress would loss of FANCM induce G2/M arrest here too (i.e. T98G)? Longterm what happens to the cells? The authors show the colony formation assays, but clearly there are still cells growing, do they eventually die?

Answer #3

This is indeed an excellent point and we thank the Reviewer for raising it. First of all, as mentioned above, we have now performed viability assays (growth curves and PI staining of non permeabilized cells) and found that FANCM depletion leads to cell death (which we also show to be PARP1-independent). We thus mention in the text that the immediate G2/M arrest is likely to be followed by PARP1-independent apoptosis or non-apoptotic cells death.

While we could not obtain T98G cells, we have tested the impact of FACM depletion on HeLa cells experiencing telomeric or generalized replication stress obtained by depleting

the shelterin factor TRF1 or by HU, respectively. Western blot and PI/FACS analysis did not reveal any further accumulation of pS33 or G2/M cells when FANCM was depleted (Fig. S4C and D). Actually, pS33 failed to accrue efficiently in HeLa cells depleted for FANCM and treated with HU (Fig. S4D), which is consistent with a role for FANCM in supporting full activation of the canonical ATR-dependent intra S-phase checkpoint⁹. Conversely, as mentioned in our Answer #1 to Reviewer #1, we now show that FANCM depletion in TRF1 over-expressing U2OS cells still leads to pS33 and G2/M accumulation, yet less severely than in control cells (Fig. 1G and H). We believe that all these new data strongly point to ALT specific functions for FANCM in maintaining telomere stability and cell proliferation and viability.

• In Figure 2C the authors state that “FANCM-mediated suppression of telomere instability is likely to be direct, because the protein was found associated with telomeric DNA in chromatin immunoprecipitation (ChIP) experiments using anti-FANCM antibodies (Fig. 2C and D)”. The argument is that FANCM is doing something distinct at ALT telomeres to mitigate replication stress and that when this function is defective this leads to genome instability and G2/M arrest. However, the data also show that FANCM binds to Alu repeats, and in fact seems to bind Alu repeats better than telomeres? How does FANCM binding to telomeric repeats compare to a known telomere binding protein, such as TRF1 or TRF2? If the argument is that FANCM binds the ALT telomeres more efficiently than non-ALT telomeres, the authors should also include ChIP with the H-ST cell line? That being said, is the result that FANCM binds Alu repeats more efficiently than telomeres accurate? What does this say about potential function of FANCM in the maintenance of Alu repeats? Do they undergo replication stress that is also mitigated by FANCM? This needs to be addressed.

Answer #4

Our ChIP experiments are only intended to show that FANCM physically associates with ALT telomeres. We do not want to imply that FANCM binds only to telomeres in ALT cells, nor that it binds to chromatin only in ALT cells. Indeed, FANCM was found to associate with chromatin fractions of several cell types⁴, and our IF experiments show a diffuse nuclear staining both in U2OS and HeLa cells (Fig. R1). The wide distribution of FANCM on chromatin is further consistent with its immunoprecipitation together with the abundant Alu repeat DNA. We believe that FANCM is associated with chromatin at large, but it might be further recruited, stabilized or activated by structures such as R-loops, which are abundant at ALT telomeres. We have clarified this throughout the text. However, we feel that studying putative functions of FANCM at Alu repeats goes beyond the scope of this study. Indeed, as shown in Fig. 2A and mentioned in the text, accumulation of pSer33 and 53BP1 outside of telomeres was negligible in ALT cells depleted for FANCM, pointing to telomeres as one of the major loci undergoing replication stress in absence of FANCM, at least within the short timeframe of our experiments (48 hours after siRNA transfection). Yet, as we now mention in the text, we do not exclude that FANCM may serve essential functions in ALT cells also at non-telomeric regions, as suggested by its physical interaction with Alu repeat DNA.

• The changes in telomeric foci size are impressive and while perhaps not in every cell they are clearly prevalent. However, the statistical quantification of the data as presented in Figure S2E,

trivializes the result given that the standard deviations are, in several cases, equal to or larger than the mean itself. Would it be useful to instead, or in addition, quantify the percentage of cells with the larger foci? I.e. % of cells with foci size greater than 70 pixels for example and include stats there? Perhaps even add that to Figure 4? It just seems like the authors are underselling this piece.

Answer #5

We fully agree that the way we presented those data was underselling. Following the suggestion of this Reviewer and of Reviewer #1 (see Answer #12 to Reviewer 1), we have now added quantifications of cells with at least 5 large foci with an area of 60 pixels or larger. Approximately 60% of FANCM-depleted cells were scored as positive, versus approximately 10% and 15% of untreated or RO-3306-treated siCt-transfected cells, respectively. Data are presented in Fig. 3F and described in the text. Please note that the original Fig. S2 is now main Fig. 3.

- *Figure 3 and 4 seem to be making a similar conclusion, that FANCM depletion leads to an increase in APB and EdU incorporation at ALT telomeres. This is not simply a product of G2/M arrest in the FANCM depleted cells as control cells arrested in G2/M with RO-3306 do not induce the same phenotypes. These two figures should be combined and condensed.*

Answer #6

We have consolidated the original figures 3 and 4 in one unique figure (current Fig. 4) showing that FANCM depletion increases several features of ALT including RAD51-containing APBs and telomeric DNA synthesis outside of S phase. Please also note that we have added IF experiments showing that POLD3 accumulates at telomeres in FANCM-depleted U2OS cells, further corroborating the idea that FANCM suppresses BIR-mediated telomeric DNA synthesis.

- *The telomere foci following FANCM depletion are significantly larger when measured by telomere FISH as compared to TRF2 immunofluorescence, is that true or potentially just the images selected?*

Answer #7

We have quantified the number of TRF2 foci and their area in FANCM-depleted U2OS cells and control cells and directly compared them with the ones of foci detected by telomeric DNA FISH. As shown in Fig. R2, it is true that the overall area of TRF2 foci is smaller than the one of telomeric foci, and the increase observed upon FANCM depletion is less important when anti-TRF2 IF is used. This might be due to decrease binding of TRF2 to telomeres in G2/M phase, decrease affinity of TRF2 to telR-loop-containing telomeres, or both. Nevertheless, please note that TRF2 is only used as a marker for telomere localization, and not to quantitatively evaluate telomere foci size (and therefore clustering), for which telomeric DNA FISH is used.

- *In Figure 5, the authors detect an increase in C-rich single stranded DNA after FANCM depletion. The results presented in 5B are quite nice, but 5A appears to demonstrate any phenotype with VA13. Perhaps VA13 just has more robust phenotypes in the absence of*

FANCM, which is fine, the authors should just include VA13 in the data presented in 5B for consistency.

Answer #8

We have performed dot-blot hybridization experiments also for VA13 and HOS cells and they are now shown together with the previous ones performed for U2OS and HeLa cells (Fig. 5B). Although less prominently than in U2OS, C-rich telomeric ssDNA accumulated also in VA13 cells depleted for FANCM. As for HeLa, no main changes in telomeric DNA were observed in FANCM-depleted HOS cells. Quantifications are shown in Fig. 5B (table below the dot-blot images) and discussed in the text.

• In Figure 7 it appears that FANCM unwind C-rich R-loops better than G-rich? Does this mean that FANCM can resolve TERRA R-loops, but perhaps not as efficiently as another factor?

Answer #9

As we previously showed, *in vitro* transcription of G-rich, TERRA-like RNA is less efficient than transcription of C-rich telomeric RNA⁷. This is why the signal from G-rich RNA in the gel shown in Fig. 7B is weaker than the one from C-rich RNA. We have now mentioned this in the text. Nevertheless, FANCM unwinds both transcripts very efficiently as shown by the complete displacement of the signal from the telR-loop plasmid position to the one of the free RNA.

• If FANCM depleted cells induce replication stress at telomeric DNA, isn't it possible that the S9.6 antibody is also detecting okazaki fragments present at stalled replication forks? So in addition to R-loops you are also detecting Okazaki fragments? Perhaps treat your DRIP experiments with RNASEH2 in addition to RNASEH1? This would further support the model that you have increased replication stress in which some is due to R-loops at telomeric DNA, but some is also due to other structures.

Answer #10

In our DRIP experiments, purified nucleic acids were treated with recombinant *E. coli* RNaseH prior to immunoprecipitation to control for specificity of the S9.6 antibody in recognizing RNA:DNA hybrid structures. We did not use RNaseH1 or RNaseH2 *in vitro*. Nevertheless, cellular over-expression of RNaseH1, which does not resolve Okazaki fragment-associated RNA:DNA hybrids, partly suppresses telomeric replication stress in FANCM depleted-cells; this supports the hypothesis that R-loops impair replication fork progression when FANCM is depleted. Nevertheless, we do not want to exclude that long RNA:DNA hybrids other than R-loops (for example double stranded RNA:DNA devoid of any displacement loop) could also accumulate upon FANCM depletion. Indeed our experiments do not allow to distinguish between ds RNA:DNA hybrids and three-stranded nucleic acids comprising an RNA:DNA hybrid and a displaced ssDNA. We now specify this in the text.

How do levels of TERRA in U2OS compare to other ALT lines, is the FANCM phenotype blunted in cells that potentially have lower TERRA expression?

Answer #11

We would like to thank the Reviewer for this very good suggestion. Indeed we previously reported that TERRA and telR-loop abundance varies among different ALT cell lines. In particular, both TERRA and telR-loops levels are higher in U2OS than in VA13 cells⁷. This might explain why FANCM depletion leads to more robust accumulation of C-rich telomeric ssDNA and C-circles in U2OS cells than in other ALT cells (Fig. 5A, B and E). We have mentioned this in the Discussion.

- *While the authors have done their diligence to include statistical analysis throughout the paper, the tests do not always seem to be appropriate for the experiment being analyzed and as a result some of the significance may be overestimated? The authors should consult a biostatistician to support their analysis.*

Answer #12

We have used paired two-tailed student's t-test and nonparametric two-tailed Mann-Whitney U test as indicated in the different figure legends. After consulting with a biostatistician, we believe that those tests are correct.

Minor revisions

- *The figures contain a multitude of abbreviations, and while they are all defined in the figure legends, it makes the figures much harder to interpret and understand.*

Answer #13

We have tried to reduce the number of abbreviations; nevertheless they have not been largely eliminated due to space constrains. Additionally, as suggested by Reviewer #3, we have indicated ALT cell lines using a grey background in all figures where more than U2OS and HeLa were used. We hope this helps the reader.

- *Figure S1 needs panel identifiers (A, B, C etc.) in order to directly reference the relevant cell cycle profile for experiments in the text.*

Answer #14

We have added panel identifiers and referenced them accordingly throughout the text.

- *In Figure 3 the authors state that FANCM depletion increases telomeric DNA synthesis outside of S-Phase, however the quantification of relative cell count for each phase of the cell cycle in Figure 3C demonstrates that there are still approximately 40-60% of cells in G1/S. How are the S-phase cells excluded from the analysis in Figure 3A-B? The cells with greater than 5-foci, but less than X foci? 10? 20? Just a little clarity on the analysis would be helpful here.*

Answer #15

Please see our Answer #8 to Reviewer #1.

- *The quantification of ECTR by FISH in Figure 5 should be better detailed. The authors should describe in words what was considered a positive ECTR event?*

Answer #16

We have now indicated in the text that we scored as ECTRs 'extrachromosomal signals positive to telomeric probe hybridization but not stained by 4',6-diamidino-2-phenylindole (DAPI)'.

- *The text incorrectly references Figure 5A as Figure 5D (sentence beginning "When hybridization was performed under native...")*

Answer #17

We have corrected this mistake.

- *In Figure 5E, it is unclear whether the U2OS and VA13 data is not significant, or whether statistics were not performed on these data.*

Answer #18

The original quantification was indeed wrong, we apologize for this. We have now included the correct quantification with statistical analysis.

- *Figure 7A lacks quantification.*

Answer #19

We have added the quantification of the northern blot shown in Fig. 7A.

- *There are not representative image for all of the conditions quantified in Figure 8B*

Answer #20

Representative images for all conditions are now shown in Fig. S7B

Reviewer #3 (Remarks to the Author):

In their manuscript entitled "FANCM assures regulated ALT activity by restricting BLM- and R-loop-induced telomeric replication stress", Penz et al have demonstrated the essential function of FANCM, specifically at telomeres that are being maintained through the ALT mechanism. Indeed they have convincingly shown that many ALT features are strongly exaggerated with FANCM loss, including c-circle accumulation, telomere replication stress and APB formation. They suggest that although replication stress is required, to an extent, for the BIR mediated repair of ALT telomeres, it is kept in check by FANCM to ensure continued viability. The authors demonstrate that the source of the replication stress is due, in part, to the accumulation of TERRA R-loops at telomeres. Accordingly, the expression of RNase H alleviates many of the siFANCM telomeric phenotypes. It is suggested, that particularly in the case of ALT cancers,

FANCM may be an effective target for cancer therapy as it seems to specifically target ALT cells and not telomerase positive cells. In conclusion, the manuscript is well written and the data depicted are solid. It is an important study that has important implications. I support its publication but do have a few points that I would like to see addressed.

Answer #1

We thank this Reviewer for his/her support and for recognizing the importance of our study.

Although the manuscript strongly suggests that the loss of viability in ALT cells is due to the mode of telomere maintenance (for good reason) it could potentially be independent of telomeres. I feel that one important control would be to test if the loss of ATRX (mutated in 95% of ALT cell lines) is synthetic lethal with loss of FANCM in telomerase positive cells.

Answer #2

This is indeed a very good point. We have now depleted ATRX and FANCM simultaneously in HeLa cells. Western blot and PI/FACS analysis did not reveal any accumulation of pS33 or G2/M cells (Fig. S4B). On the other side, as mentioned in our Answer #1 to Reviewer #1, we now show that FANCM depletion in TRF1 over-expressing U2OS cells still leads to pS33 and G2/M accumulation, yet less severely than in control cells (Fig. 1G and H). Because TRF1 over-expression alleviates replication stress in U2OS cells¹, we believe that these new data indicate that the defects in cell proliferation inflicted by FANCM depletion derive at least in part from dysfunctional telomeres.

I was surprised to see that the expression of “super telomerase” did not rescue the lethality of U2OS cells, which would suggest that the ALT telomerase maintenance mode might not be responsible for this genetic interaction. Can the authors please comment (or demonstrate) what actually happens when telomerase is overexpressed in ALT cells on the level of c-circles, APBs and TERRA levels/R-loops. Doesn't the “telomerase mode” of elongation take over, as is the case in the budding yeast system, and how can we be sure that telomerase is actually acting on telomeres in the U2OS cells.

Answer #3

We have previously shown that supertelomerase U2OS cells do not acquire a ‘telomerase mode’ of telomere elongation, as several ALT features (APBs, C-circles, elevated TERRA levels, TERRA accumulation in APBs, telomere fragility) remain largely unaltered. The only feature that is clearly affected is telomere free ends (TFEs), which are diminished in supertelomerase U2OS cells as compared to U2OS cells. This indicates that telomerase is able to elongate the shortest telomeres in ALT cells⁷ and is consistent with reports from other laboratories including the Reddel laboratory¹⁰. Thus, telomerase expression does not suppress ALT but rather the two modes of elongation coexist. We have now mentioned this in the text and we show examples of fragile telomeres and TFEs in supertelomerase U2OS cells (Fig. S3B), functionally validating that telomerase is active in our supertelomerase cells.

It is suggested that FANCM inhibition may be beneficial for ALT related cancer therapy based

on the fact that the inhibition of FANCM was not toxic in telomerase positive cells. However, since 80% of the cells in humans are telomerase negative, what are the effects in telomerase negative cells? Or in cells where telomerase has been pharmacologically or genetically inhibited?

Answer #4

As mentioned in our Answer #2 to Reviewer #2, we have now performed FANCM depletion experiments using lentivirus-mediated stable expression of shRNAs. As shown in Fig. R4, shRNA-mediated depletion was not as efficient as with siRNAs (compare depletion in U2OS in Fig. R4A with the ones showed in the manuscript). Similarly, FANCM depletion-associated defects (pS33 and G2/M cell accumulation) were milder in shRNA-infected U2OS cells than in siRNA-transfected ones (compare pS33 western blot in Fig. R4A and (PI)/FACS profiles in Fig. R4B with the ones showed in the manuscript). We have nonetheless shRNA-depleted FANCM in MRC5 and HLF primary lung fibroblasts and did not detect any accumulation of pS33 or G2/M cells (Fig. R4). This suggests that FANCM is not necessary for normal proliferation of unchallenged primary fibroblasts. Nevertheless, because of the decreased depletion efficiency observed in U2OS cells, we would prefer not to include these data in the manuscript. However, as mentioned in our Discussion section, familial FANCM loss-of-function mutations have been reported and homozygous carriers developed till adulthood although they were diagnosed with early onset cancer^{11,12}. Also, mouse knocked-out for FANCM have been successfully generated and reached adulthood¹³. It seems therefore likely that primary cells do not require FANCM for normal proliferation unless challenged with DNA damaging agents.

As for telomerase inhibition, we have now siRNA-depleted FANCM in HeLa cells treated with the telomerase inhibitor BIBR 1532 (Fig. S4A). No accumulation of pS33 and G2/M cells was detected, arguing against a protective role of telomerase against FANCM inhibition.

Minor comments:

In general, in a many of the figures it would help if the ALT cell lines and Telomerase (+) cell lines were indicated directly on the figure in some manner, maybe it's not required in all figures but at least in figure 1. Moreover, in the discussion it would be useful to refer to the relevant figures.

Answer #5

We have indicated ALT cell lines using a grey background in all figures where more than U2OS and HeLa were used. We have also referred to relevant figures throughout the Discussion.

p.9: 'Figure 5D, upper panel'. There is no upper panel for 5D. It should probably say '5A'

Answer #6

We have corrected this mistake.

'These results establish that FANCM and BLM depletion are not synthetically lethal in ALT cells, and rather reveal a synthetic viable interaction'. I think this should be toned down

accordingly as I feel that synthetic lethals are difficult to conclude with incomplete knock down experiments using siRNA.

Answer #7

We do agree with this statement. We now write: ‘These results establish that BLM depletion alleviates the adverse effects exerted by FANCM deficiency on ALT cells’.

p.12: The FISH applied in 7D could be more clearly explained. Maybe a cartoon would help to demonstrate why more C-strand is accessible to the probe following RNase H treatment.

Answer #8

We have added a cartoon to the figure.

p.13 top: ‘we propose that FANCM directly resolves telR-loops on telomeric chromatin’. This does not explain why single stranded C-rich DNA accumulates in siFa cells in the absence of RNaseH (Figure 7D and E). Could this be due to replication gaps?

Answer #9

We fully agree with this statement. We now write: ‘The more prominent C-rich ssDNA signal already present in FANCM-depleted cells not treated with RNaseH (Fig. 7D and E) might originate from gaps in DNA replication or cellular degradation of the RNA moiety of telR-loops’.

In the discussion it could be mentioned that the Aguilera lab has recently demonstrated that Mph1 is recruited to short telomeres in an RNase H sensitive manner (Lafuente-Barquero et al, 2017).

Answer #10

This is indeed a very relevant study. We now refer to it in the Discussion and we write: ‘It is conceivable that telR-loops directly promote recruitment and/or stabilization and in turn activation of FANCM at telomeres in human ALT cells, thus regulating POLD3-dependent, BIR-mediated telomere elongation’.

References

1. Lee, Y.W., Arora, R., Wischnewski, H. & Azzalin, C.M. TRF1 participates in chromosome end protection by averting TRF2-dependent telomeric R loops. *Nat Struct Mol Biol* **25**, 147-153 (2018).
2. Schwab, R.A. et al. The Fanconi Anemia Pathway Maintains Genome Stability by Coordinating Replication and Transcription. *Mol Cell* **60**, 351-61 (2015).
3. Pan, X. et al. FANCM, BRCA1, and BLM cooperatively resolve the replication stress at the ALT telomeres. *Proc Natl Acad Sci U S A* **114**, E5940-E5949 (2017).
4. Kim, J.M., Kee, Y., Gurtan, A. & D'Andrea, A.D. Cell cycle-dependent chromatin loading of the Fanconi anemia core complex by FANCM/FAAP24. *Blood* **111**, 5215-22 (2008).

5. Lafuente-Barquero, J. et al. The Smc5/6 complex regulates the yeast Mph1 helicase at RNA-DNA hybrid-mediated DNA damage. *PLoS Genet* **13**, e1007136 (2017).
6. Cristofari, G. & Lingner, J. Telomere length homeostasis requires that telomerase levels are limiting. *EMBO J* **25**, 565-74 (2006).
7. Arora, R. et al. RNaseH1 regulates TERRA-telomeric DNA hybrids and telomere maintenance in ALT tumour cells. *Nat Commun* **5**, 5220 (2014).
8. Sobinoff, A.P. et al. BLM and SLX4 play opposing roles in recombination-dependent replication at human telomeres. *EMBO J* **36**, 2907-2919 (2017).
9. Collis, S.J. et al. FANCM and FAAP24 function in ATR-mediated checkpoint signaling independently of the Fanconi anemia core complex. *Mol Cell* **32**, 313-24 (2008).
10. Perrem, K., Colgin, L.M., Neumann, A.A., Yeager, T.R. & Reddel, R.R. Coexistence of alternative lengthening of telomeres and telomerase in hTERT-transfected GM847 cells. *Mol Cell Biol* **21**, 3862-75 (2001).
11. Catucci, I. et al. Individuals with FANCM biallelic mutations do not develop Fanconi anemia, but show risk for breast cancer, chemotherapy toxicity and may display chromosome fragility. *Genet Med* **20**, 452-457 (2018).
12. Meetej, A.R. et al. A human ortholog of archaeal DNA repair protein Hef is defective in Fanconi anemia complementation group M. *Nat Genet* **37**, 958-63 (2005).
13. Bakker, S.T. et al. Fancm-deficient mice reveal unique features of Fanconi anemia complementation group M. *Hum Mol Genet* **18**, 3484-95 (2009).

Reviewers' comments:

Reviewer #1 (Remarks to the Author):

My questions have been addressed. Nice job. thank you.

Reviewer #2 (Remarks to the Author):

In this resubmission, the authors have made a number of revisions to the manuscript and performed a number of experiments to directly address the reviewers' concerns. There are still some minor concerns/comments that I think the authors could address to further improve the manuscript, but that wouldn't require additional experiments. These are listed below. Overall, this manuscript has reported some novel findings with respect to TERRA that not only deepen the mechanistic understanding of ALT telomere maintenance, but also add to the findings presented in the manuscript that was co-submitted from the Pickett lab.

- The quantification provided in Figure 5B should go into the supplement
- In Figure 5E the C-circle signal in the dot blot for the HuO9 and the U2OS look the same yet the U2OS have almost 20-fold higher signal in the quantification. These dot blot images should be more reflective of the quantification.
- The quantification of ECTR by metaphase spread does not seem like a rigorous analysis or entirely consistent with Figure 5B, D, and E. For example, the VA13 cells appear to have the same amount of ECTR as U2OS by metaphase spread, yet dot blot and C-circle assays paint a much more modest picture. One FISH positive, DAPI negative dot represents one ECTR seems like it could easily overestimate ECTR abundance and not be entirely accurate. The C-circle data is quite nice and coupled with the results in Figure 5B seems to support the conclusion that there is an increase in ECTR, but just that the abundance varies by cell line. The ECTR data from the metaphase spreads just isn't necessary nor overly compelling.
- The author states that 'FANCM and BLM co-depletion resulted in a partial rescue of the aberrant cell cycle distribution and cell proliferation and viability deriving from depleting FANCM' and then later on notes that 'MYC-RH1 WT alleviated the G2/M arrest defect in cells co-depleted for FANCM and BLM'. The wording here is confusing and should be clarified. Does the MYC-RH1 WT further enhance the rescue already partially observed in the FANCM/BLM co-depleted cells?
- Using a Mann Whitney or T-Test for statistical analysis in the bulk of this manuscript is inappropriate. A Mann Whitney is a 'nonparametric test to compare two unpaired groups to compute a P-Value'. While the authors are in fact comparing two groups in each of their analyses the bigger picture is looking at many comparisons, for example Figure 8F. The authors should consider an ANNOVA followed by a post hoc analysis.

Reviewer #3 (Remarks to the Author):

The concerns that I have raised were all addressed in a very satisfying manner. congratulations on this nice manuscript.

Point-by-point response to the Reviewer #2's comments (in blue italics).

- *The quantification provided in Figure 5B should go into the supplement*

We believe that this quantification is important, as it shows that FANCM depletion in ALT cells leads to accumulation of C-rich telomeric DNA. We do not think that moving this to supplemental material would favor the reader and we therefore prefer to keep it as part of main Figure 5. To help the reader focus on the main message of this experiment, we have now used a thicker border to highlight the part of the table showing the quantification of telomeric C-rich ssDNA in ALT cells transfected with siRNA against FANCM and with control siRNAs. This is also mentioned in the figure legend.

- *In Figure 5E the C-circle signal in the dot blot for the HuO9 and the U2OS look the same yet the U2OS have almost 20-fold higher signal in the quantification. These dot blot images should be more reflective of the quantification.*

This is true, thanks for noticing it. We have now used images that are more representative of the quantification. Please also note that the quantification shows relative values as the signal in FANCM depleted cells is expressed relative to siCt-transfected samples.

- *The quantification of ECTR by metaphase spread does not seem like a rigorous analysis or entirely consistent with Figure 5B, D, and E. For example, the VA13 cells appear to have the same amount of ECTR as U2OS by metaphase spread, yet dot blot and C-circle assays paint a much more modest picture. One FISH positive, DAPI negative dot represents one ECTR seems like it could easily overestimate ECTR abundance and not be entirely accurate. The C-circle data is quite nice and coupled with the results in Figure 5B seems to support the conclusion that there is an increase in ECTR, but just that the abundance varies by cell line. The ECTR data from the metaphase spreads just isn't necessary nor overly compelling.*

We agree that a quantitative analysis of ECTRs detected in *in situ* hybridization experiments might be misleading, as we cannot tell apart different types of ECTRs (C-circles, G-circles, T-circles and other not circular forms). We have removed the data (FISH images plus quantification) from main Figure 5. We then refer to Supplementary Figure S5, where FISH examples are shown yet without quantifications, and mention that *in situ* hybridization experiments revealed abundant extrachromosomal telomeric signals, which probably correspond to ECTRs.

- *The author states that ‘FANCM and BLM co-depletion resulted in a partial rescue of the aberrant cell cycle distribution and cell proliferation and viability deriving from depleting FANCM’ and then later on notes that ‘MYC-RH1 WT alleviated the G2/M arrest defect in cells co-depleted for FANCM and BLM’. The wording here is confusing and should be clarified. Does the MYC-RH1 WT further enhance the rescue already partially observed in the FANCM/BLM co-depleted cells?*

We have rephrased the section and we now say: ‘MYC-RH1 WT further enhanced the rescue of G2/M arrest defect in cells co-depleted for FANCM and BLM (Fig. 8E; Fig. S1E)’.

- *Using a Mann Whitney or T-Test for statistical analysis in the bulk of this manuscript is inappropriate. A Mann Whitney is a ‘nonparametric test to compare two unpaired groups to compute a P-Value’. While the authors are in fact comparing two groups in each of their analyses the bigger picture is looking at many comparisons, for example Figure 8F. The authors should consider an ANNOVA followed by a post hoc analysis.*

We agree with this point. For all statistical analyses of data comparing more than two factors for each group and their interaction we used a two-way analysis of variance (ANOVA) followed by Tukey’s HSD for the pairwise comparisons. The analysis was carried out using the aov and TukeyHSD functions of R version 3.3.2. The significance levels indicated are from the Tukey’s HSD adjusted p-values. This new analysis applies to main figures 6C, 6F, 8C and 8F as well as to supplementary figure S6C. The statistical analysis method is indicated in figure legends and in the Materials and Methods section.

REVIEWERS' COMMENTS:

Reviewer #2 (Remarks to the Author):

The revisions look great. Nice story, thank you.